# MD-RE: A Multi-Discrimination Framework for Document-Level Relation Extraction with Adaptive Threshold Shifted Loss

## Abstract

Document-level relation extraction (DocRE) aims to identify relations for an entity pair within a document. Existing methods can be broadly classified into two categories: direct encoding of the entire document or enhancement using extracted evidence sentences. However, the former often introduces noise unrelated to relations, while the latter is heavily dependent on the quality of evidence extraction. Moreover, these DocRE models typically use an adaptive threshold to predict all potential relations for an entity pair. As a result, class imbalance in DocRE often leads the model to learn a high threshold for an entity pair, which in turn causes the model to frequently predict that the entity pair has no relation. To address these issues, we propose a **M**ulti-**D**iscrimination framework (**MD-RE**) that does not rely on evidence sentences. MD-RE employs three discriminators with dynamically adjusted thresholds to independently predict relations, and aggregates their outputs via a weighted fusion strategy. Furthermore, we propose an **A**daptive **T**hreshold **S**hifted **L**oss (**ATSL**), which encourages lower threshold to alleviate the high false negative rate resulting from class imbalance. Experiments on three datasets demonstrate that our MD-RE framework with ATSL achieves new state-of-the-art results. Moreover, ATSL significantly improves the performance of various existing DocRE models. In addition, combining other losses with MD-RE also yields competitive results. Our code is available at https://anonymous.4open.science/r/MD-RE.

## 1 Introduction

Document-level relation extraction (DocRE) aims to extract relations for an entity pair from multiple sentences within a document. Since entities may span multiple sentences, the model needs reason over more complex contexts and handle more relation types, making DocRE more challenging than sentence-level relation extraction. DocRE supports tasks such as knowledge graph Mondal et al. (2021), information retrieval Zeng et al. (2024), and question answering Liu et al. (2024).

Most existing DocRE models are based on Transformer or graph-based architectures. Representative methods, such as ATLOP Zhou et al. (2021), employ localized context pooling to guide attention toward relation-relevant regions, while KD-DocRE Tan et al. (2022a) enhances multi-hop relation modeling via axial attention. In addition, some graph-based methods construct graphs and use graph neural networks to reason about relations between entities (Peng et al., 2022; Sun et al., 2023). Most of these models use the entire document as input context, but studies (Huang et al., 2021a;b) suggest that this may introduce noise irrelevant to relations. To mitigate this issue, recent works (Xie et al., 2022; Ma et al., 2023; Lu et al., 2023) propose extracting evidence sentences relevant to a given entity pair. However, these methods depend on the quality of evidence extraction and often exhibit limited effectiveness in low-resource scenarios (e.g., without evidence annotations).

To address these challenges, we propose a **M**ulti-**D**iscrimination framework (**MD-RE**), which does not rely on evidence sentences and reduces noise from the entire document through multi-discrimination perspectives. Specifically, MD-RE employs three discriminators with varying recall rates, each adopting a different threshold to determine the existence of relations for an entity pair. Higher-recall discriminators apply lower thresholds to retain more candidate relations, whereas

lower-recall ones use higher thresholds to filter them more strictly. To enable each discriminator to use a different threshold for an entity pair, we propose a **L**oss-aware **N**egative **S**election (LNS) method: for each batch, we retain all positive examples[1] and select the top-k negative examples based on their loss. By reducing the number of negative examples, *we can initially adjust the threshold and recall rate of each discriminator*.

Furthermore, in order to more flexibly adjust the threshold and recall rate of each discriminator by introducing different threshold biases, we propose a novel **A**daptive **T**hreshold **S**hifted **L**oss (ATSL). The existing DocRE models usually employ an adaptive threshold loss (ATL) Zhou et al. (2021) to predict relations for an entity pair, where a relation exists if its predicted logit exceeds the threshold. In the meantime, real-world datasets often face class imbalance[2]. This imbalance drives the model to learn a higher threshold for an entity pair, which in turn causes the model to frequently predict that the entity pair has no relation. An intuitive method is to lower the threshold at the initial stage of prediction, allowing more entity pairs to be classified as having relations.

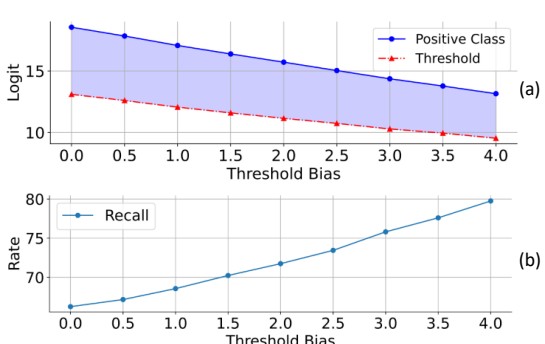

Figure 1: (a) Impact of ATSL threshold bias on the logits of the threshold and the positive class. (b) Recall changes across different threshold bias of ATSL.

*Motivated by this idea*, we propose the ATSL, which introduces threshold biases based on the ATL loss, and improves the recall rate by lowering the threshold, thereby reducing the high false negative rate caused by class imbalance. In addition, we further observe that ATSL loss has *two key capabilities* that enable the discriminators to flexibly control both thresholds and recall rates: 1) ATSL can flexibly adjust the boundary between the positive class and the threshold, giving the model a more fine-grained logit judgment ability. In **Fig. 1(a)**, as the threshold bias increases, both the threshold and the positive class logit decrease, while the boundary between the positive class and the threshold shrinks; 2) ATSL also enables flexible control of the recall rate. In **Fig. 1(b)**, increasing the threshold bias leads to an increase in recall. The contributions of our work are as follows:

- We propose a Multi-Discrimination framework (MD-RE) for DocRE, consisting of three discriminators with different discrimination criteria. Unlike previous methods, MD-RE framework does not rely on evidence sentences and effectively reduces document-level noise by incorporating multiple discrimination perspectives.
- We propose a Loss-aware Negative Selection (LNS) method to initially adjust the threshold and recall rate of each discriminator, and design a weighted fusion strategy to aggregate their outputs, aiming to achieve better prediction performance.
- We propose ATSL, a novel loss that introduces threshold biases to more flexibly adjust the threshold and recall rate of each discriminator and effectively mitigate class imbalance.
- Results on three datasets show that our MD-RE framework with ATSL loss achieves state-of-the-art performance. Notably, **MD-RE framework and ATSL loss are independent yet complementary, each capable of improving DocRE**. ATSL loss consistently enhances performance and generalizes well across different baselines (average +2.52 F1). Moreover, even without ATSL loss, MD-RE framework with only ATL loss still outperforms the baseline model by +3.04 F1.

## 2 RELATED WORK

**Document-Level Relation Extraction.** DocRE methods can be broadly categorized into: (1) directly encoding the entire document, such as GAIN Zeng et al. (2020), ATLOP Zhou et al. (2021), DocuNet Zhang et al. (2021), KMGRE Jiang et al. (2022), KD-DocRE Tan et al. (2022a), TTM-RE Gao et al. (2024), ABRE Xu et al. (2024), and VaeDiff-DocRE Tran et al. (2025); (2) introducing

---

[1]An entity pair is a *positive example* if it has at least one relation; otherwise, it's a *negative example*.

[2]Class imbalance: negative examples significantly outnumber positive ones. For instance, in the Re-DocRED Tan et al. (2022b) dataset, about 94% of entity pairs have no relation.

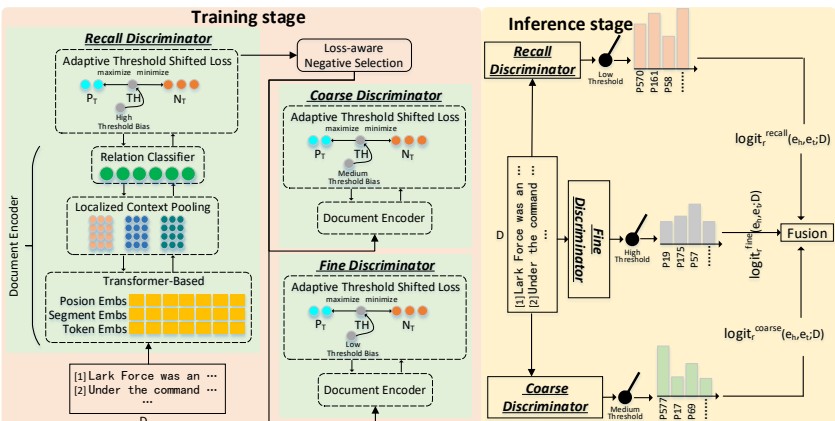

Figure 2: Overview of our MD-RE framework. In training, three discriminators with distinct decision criteria apply different thresholds to determine whether a relation exists for an entity pair. The combination of Loss-aware Negative Selection (LNS) and Adaptive Threshold Shifted Loss (ATSL) enables more effective dynamic adjustment of the threshold and recall for each discriminator. In inference, we adopt a weighted fusion strategy to integrate the outputs of three discriminators.

evidence sentences, including Eider Xie et al. (2022), SAIS Xiao et al. (2022), DREEAM Ma et al. (2023), and AA Lu et al. (2023). In addition, due to false negatives in DocRED Yao et al. (2019), we evaluate robustness by testing on the revised version, Re-DocRED Tan et al. (2022b), after training on DocRED. Effective methods include SSR-PU Wang et al. (2022), CAST Tan et al. (2023), and P³M Wang et al. (2024).

**Loss for DocRE.** DocRE typically employs the ATL Zhou et al. (2021) loss, which adaptively assigns a threshold to each entity pair, considering a relation to exist only when its predicted logit surpasses the threshold. Based on this, Tan et al. (2022a); Zhou & Lee (2022) find that the class imbalance is prevalent in DocRE. To address this, some subsequent studies have enhanced ATL, including Balanced-Softmax Zhang et al. (2021), AML Wei & Li (2022), AFL Tan et al. (2022a), SSR-PU Wang et al. (2022), NCRL Zhou & Lee (2022), PEMSCL Guo et al. (2023), HingeABL Wang et al. (2023) and CMM Duan et al. (2025). The above losses mitigate class imbalance via decision boundary optimization but lack flexible threshold and recall adjustment, limiting their effectiveness. To this end, we propose the **A**daptive **T**hreshold **S**hifted **L**oss (**ATSL**).

## 3 METHODOLOGY

Our MD-RE framework in **Fig. 2** consists of four main parts: Document Encoding module, Discrimination and Loss-aware Negative Selection module, Fusion module, and our loss ATSL.

### 3.1 PROBLEM FORMULATION

The goal of DocRE is to predict relations $R \cup \{\text{NA}\}$ for entity pairs $(e_h, e_t)_{h,t=1}^n, h \neq t$ within a document $D$. Here, $\{e_i\}_{i=1}^n$ denotes the set of entities in the document, and $e_h$ and $e_t$ refer to the head and tail entities, respectively. $R$ is a predefined set of relations, while NA indicates the absence of any relation. For an entity pair $T=(e_h, e_t)$, the positive classes $\mathcal{P}_T \subseteq R$ correspond to relations expressed by any entity mention pair, whereas the negative classes $\mathcal{N}_T \subseteq R$ represent relations not expressed between them. If $T$ expresses no relations, $\mathcal{P}_T$ is empty, and $\mathcal{N}_T = R$.

### 3.2 DOCUMENT ENCODING MODULE

The token sequence of a document $D$ is denoted as $\text{T}_D = \{t_i\}_{i=1}^{|\text{T}_D|}$, where a special token "*" is inserted at the beginning and end of each entity mention. Following ATLOP Zhou et al. (2021) and DREEAM Ma et al. (2023), we obtain token-level hidden states $\text{H} \in \mathbb{R}^{|\text{T}_D| \times d}$ and attention weights $\text{A} \in \mathbb{R}^{|\text{T}_D| \times |\text{T}_D|}$ by averaging outputs and last-head attentions from the last three encoder layers,

respectively, where $d$ is the hidden size:

$$H, A = PLM(T_D) \tag{1}$$

The embedding $h_e$ for each entity $e$ is obtained by aggregating information from all its mentions $M_e = \{m_i\}_{i=1}^{|M_e|}$, where $H_{m_i}$ denotes the embedding of the special token "*" that marks the starting position of the $i$-th mention:

$$h_e = \log \sum_{i=1}^{|M_e|} \exp(H_{m_i}) \tag{2}$$

Then we use the localized context pooling method to compute $c_{h,t}$ from token embeddings $H$ and cross-token attention $A$, where $A_h$ and $A_t$ represent attention for entities $e_h$ and $e_t$, and $\otimes$ denotes element-wise product.

$$c_{h,t} = H^\top \frac{A_h \otimes A_t}{A_h^\top A_t} \tag{3}$$

The localized context $c_{h,t}$ is concatenated with the $h_{e_h}$ and $h_{e_t}$ individually. Here, $\|$ denotes concatenation, $W_h, W_t$ are trainable weights, and $b_h, b_t$ are the biases for head and tail entities. Finally, $z_h$ and $z_t$ are fed into a bilinear classifier to compute the relation logits $logit_{h,t}$ for the $(e_h, e_t)$:

$$\begin{aligned} z_h &= \tanh(W_h[h_{e_h}\|c_{h,t}] + b_h) \\ z_t &= \tanh(W_t[h_{e_t}\|c_{h,t}] + b_t) \end{aligned} \tag{4}$$

$$logit_{h,t} = z_h^\top W_r z_t + b_r \tag{5}$$

### 3.3 Discrimination and Loss-aware Negative Selection Module

The core idea of our MD-RE framework is to progressively refine candidate relations through multiple stages. To achieve this, we employ three discriminators with distinct criteria, each applying a differently adjustable threshold to determine whether relations exist between an entity pair. The *motivation for designing the three discriminators and the details of each one* are described in the corresponding sections below.

**Recall Discriminator.** This discriminator is designed to achieve a high recall rate by applying lower thresholds to retain more candidate relations. Specifically, given an entity pair, we obtain its $logit_{h,t}$ (see **Eq. (5)**), and then evaluate it using our ATSL loss (see **Section 3.5**). By adjusting the hyperparameter $\lambda$ of ATSL, we can flexibly control the threshold and recall rate of the discriminator.

**Loss-aware Negative Selection.** After the recall discriminator, there are still a large number of negative samples, most of which are easy for the model to classify. This causes the model to focus on trivial instances while overlooking harder, more informative ones. To address this, we introduce the **L**oss-aware **N**egative **S**election (**LNS**) method. By applying LNS, we effectively reduce the impact of too many negative samples, which helps to improve the performance of subsequent coarse and fine discriminators. In addition, *LNS can also cooperate with ATSL loss to further dynamically adjust the threshold and recall of each discriminator.* Specifically, for each batch, we retain all positive examples and select the Top-$k$ negative examples based on their loss, defined as:

$$\begin{aligned} k &= \min(\rho \cdot |\mathcal{S}_{pos}|, |\mathcal{S}_{neg}|) \\ \mathcal{S}_{neg\text{-}hard} &= \text{Top-}k\,(\mathcal{S}_{neg}) \end{aligned} \tag{6}$$

Here, $\mathcal{S}_{pos}$ and $\mathcal{S}_{neg}$ denote the sets of positive and negative examples within the batch, respectively. $\rho$ controls the ratio of selected negatives relative to the number of positives.

**Coarse Discriminator.** Subsequently, we feed all positive samples along with the negative samples selected by the LNS method into the coarse discriminator, and incorporate the ATSL loss to dynamically set a moderate threshold for each entity pair. This discriminator aims to further filter candidate relations at a coarse granularity.

**Fine Discriminator.** The fine discriminator is trained similarly to the coarse discriminator but with a higher threshold. It applies stricter criteria than the coarse discriminator to further refine candidate relations and improve prediction reliability.

## 3.4 Fusion Module

Since the three discriminators use different decision criteria, we design a weighted fusion strategy to effectively integrate their outputs and make the final decision, fully leveraging their respective strengths. Specifically, if the recall discriminator predicts that a triplet $(h, r, t)$ has the NA, indicating the absence of a relation, we accept this prediction. Otherwise, if all three discriminators predict the existence of a triplet $(h, r, t)$, we directly accept the prediction; if not, we compute a fused logit, where $\text{logit}_{h,r,t}^{\text{recall}}$, $\text{logit}_{h,r,t}^{\text{coarse}}$, and $\text{logit}_{h,r,t}^{\text{fine}}$ denote the relation logit of $r$ predicted by three discriminators for an entity pair.

$$\text{logit}_{h,r,t}^{\text{final}} = \text{logit}_{h,r,t}^{\text{recall}} + \text{logit}_{h,r,t}^{\text{coarse}} + \text{logit}_{h,r,t}^{\text{fine}} \tag{7}$$

We further define the final adaptive threshold based on the threshold of each discriminator. Similarly, $\text{logit}_{h,\text{TH},t}^{\text{recall}}$, $\text{logit}_{h,\text{TH},t}^{\text{coarse}}$, and $\text{logit}_{h,\text{TH},t}^{\text{fine}}$ represent the threshold logits of the three discriminators.

$$\text{logit}_{h,\text{TH},t}^{\text{final}} = \alpha \cdot \text{logit}_{h,\text{TH},t}^{\text{recall}} + \text{logit}_{h,\text{TH},t}^{\text{coarse}} + \text{logit}_{h,\text{TH},t}^{\text{fine}} \tag{8}$$

A relation $r$ exists if $\text{logit}_{h,r,t}^{\text{final}} > \text{logit}_{h,\text{TH},t}^{\text{final}}$. The method leverages complementary decision patterns of discriminators to improve performance. Since the recall discriminator plays a more dominant role in controlling recall, we apply the weighting factor $\alpha$ only to its threshold.

## 3.5 ATSL Loss Design

**An Empirical Analysis of ATL.** As shown in **Eq. (9)**, the Adaptive Threshold Loss (ATL) Zhou et al. (2021) divides the set $R$ of predefined relations into two subsets: the positive classes $\mathcal{P}_T$ and the negative classes $\mathcal{N}_T$, with an external threshold class TH used to distinguish between them. The objective is to encourage the logits of $\mathcal{P}_T$ to be higher than that of the TH class, and the logits of $\mathcal{N}_T$ to be lower than that of the TH class.

$$\mathcal{L}_1 = -\sum_{r \in \mathcal{P}_T} \log\left(\frac{\exp(\text{logit}_r)}{\sum_{r' \in \mathcal{P}_T \cup \{\text{TH}\}} \exp(\text{logit}_{r'})}\right)$$

$$\mathcal{L}_2 = -\log\left(\frac{\exp(\text{logit}_{\text{TH}})}{\sum_{r' \in \mathcal{N}_T \cup \{\text{TH}\}} \exp(\text{logit}_{r'})}\right) \tag{9}$$

$$\mathcal{L}_{ATL} = \mathcal{L}_1 + \mathcal{L}_2$$

Wang et al. (2023)'s analysis finds a significant difference in the number of relations between $\mathcal{P}_T$ and $\mathcal{N}_T$, with $\mathcal{N}_T$ being larger, causing $\mathcal{L}_2$ to dominate the loss calculation. Building on this analysis, we find that the dominance of $\mathcal{L}_2$ stems not only from the imbalance between $\mathcal{P}_T$ and $\mathcal{N}_T$, but more fundamentally from the overwhelming proportion of negative examples. When no relation exists between an entity pair, $\mathcal{P}_T$ is empty, resulting in no contribution from $\mathcal{L}_1$, and the loss is solely determined by $\mathcal{L}_2$. Since most entity pairs have no relation, $\mathcal{L}_2$ dominates. Building on this and Wang et al. (2023), we reformulate $\mathcal{L}_2$ as in **Eq. (10)**. When $\text{logit}_{r'} - \text{logit}_{\text{TH}} \to -\infty$, $\mathcal{L}_2 \to 0$, indicating that $\text{logit}_{\text{TH}} \gg \text{logit}_{r'}$. This implies ATL assigns relatively high thresholds to entity pairs

$$\mathcal{L}_2 = -\log\left(\frac{\exp(\text{logit}_{\text{TH}})}{\sum_{r' \in \mathcal{N}_T \cup \{\text{TH}\}} \exp(\text{logit}_{r'})}\right)$$

$$= -\log\left(\frac{1}{1 + \sum_{r' \in \mathcal{N}_T} \exp(\text{logit}_{r'} - \text{logit}_{\text{TH}})}\right) \tag{10}$$

**Adaptive Threshold Shifted Loss.** As analyzed above, we extend the findings of Wang et al. (2023) and further reveal the limitations: when the number of entity pairs with no relations significantly exceeds the number of entity pairs with relations (class imbalance), the logit of threshold class TH increases and eventually surpasses the logits of many candidate relations, leading to a large number of false negative predictions. To address this issue, we introduce a threshold bias $\lambda > 0$ in the TH class to ensure that:

$$\mathcal{L}_2' = -\log\left(\frac{\exp(\text{logit}_{\text{TH}} + \lambda)}{\exp(\text{logit}_{\text{TH}} + \lambda) + \sum_{r' \in \mathcal{N}_T} \exp(\text{logit}_{r'})}\right)$$

$$= -\log\left(\frac{1}{1 + \sum_{r' \in \mathcal{N}_T} \exp(\text{logit}_{r'} - (\text{logit}_{\text{TH}} + \lambda))}\right) \tag{11}$$

Minimizing $\mathcal{L}_2'$ requires that:

$$\text{logit}_{r'} - (\text{logit}_{\text{TH}} + \lambda) \to -\infty \tag{12}$$

Consequently, we have:

$$\begin{aligned}\text{logit}_{\text{TH}} + \lambda &\gg \text{logit}_{r'} \\ \text{logit}_{\text{TH}} &\gg \text{logit}_{r'} - \lambda\end{aligned} \tag{13}$$

This shows that $\lambda$ effectively reduces the $\text{logit}_{\text{TH}}$ in the optimization process. From **Eq. 13**, adding $\lambda$ to the target logit boosts its value in the softmax calculation, allowing the model to achieve the same margin with a smaller $\text{logit}_{\text{TH}}$.

Similarly, we add a threshold bias $\beta$ to the other part of loss $\mathcal{L}_1'$, as shown in **Eq. (14)**. When $\mathcal{L}_1' \to 0$, it implies that $\text{logit}_{\text{TH}} + \beta - \text{logit}_r \to -\infty$. From this, we can derive that $\text{logit}_{\text{TH}} + \beta \ll \text{logit}_r$.

$$\begin{aligned}\mathcal{L}_1' &= -\sum_{r \in \mathcal{P}_T} \log\left(\frac{\exp(\text{logit}_r)}{\exp(\text{logit}_{\text{TH}} + \beta) + \sum_{r' \in \mathcal{P}_T}\exp(\text{logit}_{r'})}\right) \\ &= -\sum_{r \in \mathcal{P}_T} \log\left(\frac{1}{1 + \exp(\text{logit}_{\text{TH}} + \beta - \text{logit}_r) + \sum_{r' \in \mathcal{P}_T, r' \neq r}\exp(\text{logit}_{r'} - \text{logit}_r)}\right)\end{aligned} \tag{14}$$

Minimizing ATSL implies the following margin constraints between logits, which ensure a positive margin between classes and theoretically improve generalization Vapnik (1998), as shown in **Eq. (15)**. **More theoretical proof of ATSL loss is detailed in Appendix A.**

$$\text{logit}_r - \text{logit}_{\text{TH}} \geq \beta, \quad \forall r \in \mathcal{P}_T, \qquad \text{logit}_{\text{TH}} - \text{logit}_{r'} \geq \lambda, \quad \forall r' \in \mathcal{N}_T. \tag{15}$$

Finally, we obtain the **A**daptive **T**hreshold **S**hifted **L**oss (**ATSL**), as shown in **Eq. (16)**.

$$\begin{aligned}\mathcal{L}_1' &= -\sum_{r \in \mathcal{P}_T}\log\left(\frac{\exp(\text{logit}_r)}{\exp(\text{logit}_{\text{TH}} + \beta) + \sum_{r' \in \mathcal{P}_T}\exp(\text{logit}_{r'})}\right) \\ \mathcal{L}_2' &= -\log\left(\frac{\exp(\text{logit}_{\text{TH}} + \lambda)}{\exp(\text{logit}_{\text{TH}} + \lambda) + \sum_{r' \in \mathcal{N}_T}\exp(\text{logit}_{r'})}\right) \\ \mathcal{L}_{ATSL} &= \mathcal{L}_1' + \mathcal{L}_2'\end{aligned} \tag{16}$$

## 4 EXPERIMENTAL SETUP

**Implementation Details.** Our experiments are implemented using PyTorch Paszke (2019) and Transformers Wolf et al. (2020), using $\text{BERT}_{\text{base}}$ Devlin et al. (2019) and $\text{RoBERTa}_{\text{large}}$ Liu et al. (2019) as encoders. See **Appendix B.1** for details.

**Datasets and Metrics.** We experiment on the DocRED Yao et al. (2019), DWIE Zaporojets et al. (2021), and Re-DocRED Tan et al. (2022b) datasets, which are detailed in **Appendix B.2**. We follow Zhou et al. (2021) and evaluate using F1 and Ign-F1, where **F1 represents** the standard F1, while **Ign-F1 is** computed by excluding relational facts shared between the train and dev/test sets.

## 5 MAIN RESULTS AND ANALYSIS

We conduct experiments to answer questions about our main contributions: MD-RE and ATSL.

- **Q1:** How does MD-RE framework with ATSL loss perform? (Section 5.1)
- **Q2:** How effective is our ATSL loss when applied to different models? (Section 5.2)
- **Q3:** How does the performance of our ATSL loss compare to other losses? (Section 5.2)
- **Q4:** An ablation study. (Section 5.3)

### 5.1 MAIN RESULTS

**Results on Re-DocRED.** As shown in **Table 1**, our MD-RE framework consistently outperforms all strong baselines and the previous SOTA models. Specifically, with the $\text{BERT}_{\text{base}}$ encoder, MD-RE

Table 1: Results on Re-DocRED. The underlined values indicate the results of the previous SOTA. † from Lu et al. (2023), * from original paper, ◇ from Zhang et al. (2023), ‡ from Tran et al. (2025), and △ our reproduced results.

| Model | Dev | | Test | |
|---|---|---|---|---|
| | F1 | Ign-F1 | F1 | Ign-F1 |
| with BERT$_{base}$ | | | | |
| ATLOP Zhou et al. (2021) | 74.22 ◇ | 73.35 ◇ | 74.02 ◇ | 73.22 ◇ |
| DocuNET Zhang et al. (2021) | 74.65 ◇ | 73.68 ◇ | 74.49 ◇ | 73.60 ◇ |
| KD-DocRE Tan et al. (2022a) | 74.69 ◇ | 73.76 ◇ | 74.55 ◇ | 73.67 ◇ |
| DREEAM Ma et al. (2023) | 74.58 △ | 73.74 △ | 74.23 △ | 73.42 △ |
| CAST Tan et al. (2023) | - | - | 74.67 ‡ | 73.32 ‡ |
| SA-KD Zhang et al. (2023) | 75.85 ◇ | 75.03 ◇ | 75.77 ◇ | 74.85 ◇ |
| ABRE Xu et al. (2024) | 76.26 * | 75.54 * | 76.30 * | 75.70 * |
| VaeDiff-DocRE Tran et al. (2025) | 75.89 ‡ | 74.96 ‡ | 75.07 ‡ | 74.13 ‡ |
| MD-RE (ours) | **77.70±0.10** | **76.46±0.07** | **77.80±0.04** | **76.63±0.02** |
| with RoBERTa$_{large}$ | | | | |
| ATLOP Zhou et al. (2021) | 77.63 † | 76.88 † | 77.73 † | 76.94 † |
| DocuNET Zhang et al. (2021) | 78.16 † | 77.53 † | 77.92 † | 77.27 † |
| KD-DocRE Tan et al. (2022a) | 78.65 † | 77.92 † | 78.35 † | 77.63 † |
| PEMSCL Guo et al. (2023) | 79.89 † | 79.02 † | 79.86 † | 79.01 † |
| AA Lu et al. (2023) | 81.15 † | 80.04 † | 81.20 † | 80.12 † |
| TTM-RE Gao et al. (2024) | 78.13 * | 78.05 * | 79.95 * | 78.20 * |
| VaeDiff-DocRE Tran et al. (2025) | 79.19 ‡ | 78.35 ‡ | 79.03 ‡ | 78.22 ‡ |
| MD-RE (ours) | **81.44±0.12** | **80.38±0.12** | **81.49±0.05** | **80.45±0.06** |

achieves F1 of 77.70 and 77.80 on the dev and test sets, respectively, outperforming the previous SOTA model ABRE by 1.44 and 1.50. Similarly, with the RoBERTa$_{large}$ encoder, MD-RE achieves F1 of 81.44 and 81.49 on the dev and test sets, respectively, outperforming the SOTA model AA by 0.29 points on both splits.

**Results on DWIE.** As shown in **Table 2**, MD-RE consistently outperforms baseline models on DWIE dataset, reaching 73.81 F1 and 68.37 Ign-F1 on the dev set, and 78.32 F1 and 71.28 Ign-F1 on the test set. Compared with strong baseline MILR, MD-RE improves F1 and Ign-F1 on the test set by 1.81 and 1.44, respectively. Notably, MD-RE also surpasses the recent DREEAM model, further demonstrating its effectiveness across challenging DocRE benchmarks.

Table 2: Results on DWIE with BERT$_{base}$. * from Jiang et al. (2022), ◇ from original paper, and ‡ from ours.

| Model | Dev | | Test | |
|---|---|---|---|---|
| | F1 | Ign-F1 | F1 | Ign-F1 |
| GAIN * | 62.55 | 58.63 | 67.57 | 62.37 |
| ATLOP * | 69.96 | 63.57 | 74.36 | 67.56 |
| KMGRE * | 71.40 | 65.56 | 76.71 | 69.94 |
| MILR ◇ | 72.05 | 67.18 | 76.51 | 69.84 |
| DREEAM ‡ | 72.40 | 65.93 | 74.66 | 67.27 |
| TTM-RE ‡ | 64.51 | 56.62 | 65.01 | 54.71 |
| MD-RE (ours) | **73.81±0.32** | **68.37±0.36** | **78.32±0.12** | **71.28±0.10** |

Table 3: Results on DocRED using RoBERTa$_{large}$. * from Wang et al. (2024); † from Tan et al. (2023).

| Model | Test | |
|---|---|---|
| | F1 | Ign-F1 |
| ATLOP Zhou et al. (2021) * | 45.19 | 45.09 |
| DocuNET Zhang et al. (2021) † | 45.99 | 45.88 |
| KD-DOcRE Tan et al. (2022a) † | 47.57 | 47.32 |
| SSR-PU Wang et al. (2022) * | 59.50 | 58.68 |
| CAST Tan et al. (2023) † | 65.32 | 64.25 |
| P³M Wang et al. (2024) * | 64.34 | 63.16 |
| MD-RE (ours) | **65.93±0.04** | **64.96±0.03** |

**Results on DocRED.** To evaluate the weakly supervised generalization ability of MD-RE, we train on the incomplete dataset DocRED and test on the fully annotated dataset Re-DocRED. **Table 3** shows that the MD-RE achieves the best results among all compared methods, with an F1 of 65.93 and an Ign-F1 of 64.96 on the test set. Compared to the previous competitive model CAST, MD-RE obtains gains of 0.61 F1 and 0.71 Ign-F1. When compared with other strong baselines such as P³M and SSR-PU, MD-RE's improvement ranges from a minimum of 1.59 to a maximum of 6.43.

## 5.2 RESULTS OF ATSL

**Different DocRE Models with ATSL.** To evaluate the generality of our loss, we *apply ATSL to different models by replacing their original losses.* ATLOP and DREEAM use ATL loss, DocuNet uses Balanced-Softmax loss, KD-DocRE uses AFL loss, and TTM-RE uses S-PU loss. **Table 4** shows that the ATSL loss significantly improves the performance of all baseline models. Specifi-

Table 4: Performance of different DocRE models using ATSL loss. * from Lu et al. (2023), △ from Zhang et al. (2023), ⋄ from original paper, and † our reproduced results. "**with ATSL**" indicates replacing the original loss with our ATSL loss.

| Model | Dev | | | | Test | | | |
|---|---|---|---|---|---|---|---|---|
| | F1 | F1 with ATSL | Ign-F1 | Ign-F1 with ATSL | F1 | F1 with ATSL | Ign-F1 | Ign-F1 with ATSL |
| Re-DocRED with BERT$_{base}$ | | | | | | | | |
| ATLOP Zhou et al. (2021) | 74.22 △ | **76.23** (+2.01) | 73.35 △ | **74.83** (+1.48) | 74.02 △ | **76.48** (+2.46) | 73.22 △ | **75.12** (+1.90) |
| DocuNet Zhang et al. (2021) | 74.65 △ | **76.26** (+1.61) | 73.68 △ | **74.81** (+1.13) | 74.49 △ | **76.45** (+1.96) | 73.60 △ | **75.07** (+1.47) |
| KD-DocRE Tan et al. (2022a) | 74.69 △ | **76.70** (+2.01) | 73.76 △ | **75.52** (+1.76) | 74.55 △ | **76.65** (+2.10) | 73.67 △ | **75.50** (+1.83) |
| DREEAM Ma et al. (2023) | 74.58 † | **76.08** (+1.50) | 73.74 † | **74.81** (+1.07) | 74.23 † | **76.14** (+1.91) | 73.42 † | **74.92** (+1.50) |
| TTM-RE Gao et al. (2024) | 76.21 † | **80.16** (+3.95) | 74.74 † | **79.05** (+4.31) | 76.33 † | **80.51** (+4.18) | 74.89 † | **79.48** (+4.59) |
| Re-DocRED with RoBERTa$_{large}$ | | | | | | | | |
| ATLOP Zhou et al. (2021) | 77.63 * | **80.35** (+2.72) | 76.88 * | **79.22** (+2.34) | 77.73 * | **80.40** (+2.67) | 76.94 * | **79.29** (+2.35) |
| DocuNet Zhang et al. (2021) | 78.16 * | **79.76** (+1.60) | 77.53 * | **78.78** (+1.25) | 77.92 * | **79.85** (+1.93) | 77.27 * | **78.91** (+1.64) |
| KD-DocRE Tan et al. (2022a) | 78.65 * | **79.06** (+0.41) | 77.92 * | **78.07** (+0.15) | 78.35 * | **78.76** (+0.41) | 77.63 * | **77.78** (+0.15) |
| DREEAM Ma et al. (2023) | 77.60 † | **79.56** (+1.96) | 77.20 † | **78.62** (+1.42) | 77.94 ⋄ | **79.86** (+1.92) | 77.34 ⋄ | **78.96** (+1.62) |
| TTM-RE Gao et al. (2024) | 78.13 ⋄ | **82.57** (+4.44) | 78.05 ⋄ | **81.70** (+3.65) | 79.95 ⋄ | **82.36** (+2.41) | 78.20 ⋄ | **81.53** (+3.33) |

cally, the TTM-RE model with BERT$_{base}$ achieves improvements of **4.18** in F1 and **4.59** in Ign-F1 on the test set. Similarly, the ATLOP model with RoBERTa$_{large}$ achieves gains of **2.67** in F1 and **2.35** in Ign-F1. Moreover, our loss achieves an average improvement of 2.52 in F1 on the BERT test set, and 2.23 in F1 on the RoBERTa dev set. These results demonstrate the generality of ATSL in enhancing different DocRE models.

**ATSL vs. Other Loss.** To verify the effectiveness of ATSL *compared to other losses*, we evaluate them using the ATLOP and BERT$_{base}$ encoder. **Table 5** shows that ATSL achieves 76.48 in F1 and 75.12 in Ign-F1, outperforming other losses. Specifically, compared to the current SOTA CMM loss, ATSL improves F1 and Ign-F1 by 0.36 each, and achieves an average improvement of **2.62** F1 over other losses. These results clearly demonstrate its effectiveness in the DocRE task.

Table 5: Results of different losses on Re-DocRED test set. * from Wang et al. (2023), † from Duan et al. (2025). Using ATLOP Zhou et al. (2021) and BERT$_{base}$ for encoding.

| Loss Function | F1 | Ign-F1 |
|---|---|---|
| ATL Zhou et al. (2021) * | 73.29 | 72.46 |
| Balanced-Softmax Zhang et al. (2021) * | 73.68 | 72.85 |
| AML Wei & Li (2022) * | 72.60 | 71.78 |
| AFL Tan et al. (2022a) * | 74.15 | 73.20 |
| HingeABL$_{SAT}$ Wang et al. (2023) * | 73.46 | 72.61 |
| HingeABL$_{MeanSAT}$ Wang et al. (2023) * | 74.68 | 72.90 |
| HingeABL Wang et al. (2023) * | 75.15 | 73.84 |
| CMM Duan et al. (2025) † | 76.12 | 74.76 |
| ATSL (Our Loss) | **76.48** (0.36↑) | **75.12** (0.36↑) |

Table 6: An ablation study on the Re-DocRED dev set using BERT$_{base}$ as the encoder, where $w/o$ denotes removal and $w$ indicates inclusion.

| Model | F1 | Ign-F1 |
|---|---|---|
| MD-RE (ours) | **77.70** | **76.46** |
| $w/o$ ATSL $w$ ATL | 77.25 | 76.18 |
| $w/o$ LNS | 77.52 | 76.27 |
| $w/o$ Weighted $w$ Pipeline | 72.83 | 72.27 |
| $w/o$ Weighted $w$ Add | 77.02 | 75.53 |
| $w/o$ Coarse Discriminator | 77.18 | 75.67 |
| $w/o$ Fine Discriminator | 76.38 | 74.58 |
| $only$ Recall Discriminator | 76.23 | 74.83 |

## 5.3 Ablation Study

We perform an ablation study to assess each component's impact. See **Table 6** for results, with further details in **Appendix C**.

$w/o$ ATSL $w$ ATL. Replacing ATSL loss with ATL loss slightly reduces performance. On the one hand, ATSL can be better combined with LNS; on the other hand, ATSL alleviates class imbalance.

$w/o$ LNS. After removing LNS, the F1 drops by 0.18, indicating that removing it does not affect MD-RE's ability to set different thresholds for individual discriminators, thanks to the ATSL loss. In addition, the negative samples selected by LNS also bring a slight performance gain.

$w/o$ Weighted $w$ Pipeline. Replacing the weighted fusion with a pipeline fusion severely reduces performance, as it only retains triplets unanimously judged as true by all three discriminators, discarding many potential relations.

$w/o$ Weighted $w$ Add. Replacing the weighted fusion with a union fusion results in a performance drop. Results indicate that the weighted fusion module effectively integrates information from different discriminators and exploits their complementary capabilities.

$w/o$ **Coarse Discriminator.** Removing the coarse discriminator results in a slight decrease of 0.52 in the F1. This suggests that the coarse discriminator brings a slight improvement.

$w/o$ **Fine Discriminator.** Removing the fine discriminator results in a slightly larger performance drop than removing the coarse one, with the F1 decreasing by 1.32. This suggests that the fine discriminator contributes more to refining the process.

*only* **Recall Discriminator.** Using only the recall discriminator leads to a drop in performance. This shows that the recall discriminator captures many candidates but lacks the refinement of the other two discriminators. Consequently, many false positives are retained, reducing overall precision.

## 6 FURTHER ANALYSIS

To further investigate our method's performance, we answer the following research questions:

- **Q5:** How does our MD-RE compare to other models under the same loss? (Section 6.1)
- **Q6:** Does the ATSL loss alleviate the class imbalance? (Section 6.2)
- **Q7:** How resource-efficient is MD-RE with ATSL compared to other models? (Section 6.3)
- **Q8:** What is the training cost of our ATSL loss? (Section 6.4)
- **Q9:** How does MD-RE framework perform on long documents? (Appendix D.1)
- **Q10:** How does the number of discriminators affect the MD-RE framework? (Appendix D.2)
- **Q11:** How does hyperparameter $\lambda$ influence the performance of ATSL loss? (Appendix D.3)
- **Q12:** How does hyperparameter $\alpha$ affect MD-RE performance with ATSL loss? (Appendix D.4)
- **Q13:** Case Study and Error Analysis. (Appendix D.5)

### 6.1 MD-RE VS. BASELINES UNDER SAME LOSSES

To verify the effectiveness and generalizability of the MD-RE framework, **Table 7** compares MD-RE with strong baselines under the same losses. Under the ATL loss, MD-RE significantly outperforms ATLOP, achieving a 3.04 and 2.80 improvement in F1 and Ign-F1, respectively. Similarly, when trained with the PEMSCL loss, MD-RE surpasses VaeDiff-DocRE by 2.33 in F1 and 1.96 in Ign-F1. Within MD-RE, these methods improve F1 by an average of 2.23. These consistent improvements across losses demonstrate MD-RE's effectiveness and generalizability.

Table 7: MD-RE vs. baselines under same losses, all using BERT$_{base}$ on the Re-DocRED test set.

| Loss + Model | F1 | Ign-F1 |
|---|---|---|
| ATL loss Zhou et al. (2021) | | |
| + ATLOP | 74.02 | 73.22 |
| + MD-RE (ours) | **77.06** (3.04↑) | **76.02** (2.80↑) |
| AFL loss Tan et al. (2022a) | | |
| + KD-DocRE | 74.55 | 73.67 |
| + MD-RE (ours) | **77.41** (2.86↑) | **76.19** (2.52↑) |
| NCRL loss Xu et al. (2024) | | |
| + ABRE | 76.30 | 75.70 |
| + MD-RE (ours) | **76.99** (0.69↑) | **75.97** (0.27↑) |
| PEMSCL loss Tran et al. (2025) | | |
| + VaeDiff-DocRE | 75.07 | 74.13 |
| + MD-RE (ours) | **77.40** (2.33↑) | **76.09** (1.96↑) |

### 6.2 ANALYZING THE CLASS IMBALANCE

To illustrate the effectiveness of our ATSL in alleviating class imbalance, we apply ATSL to various models and examine false negatives (FN), false positives (FP), and AUC. As shown in Table 8, ATSL substantially reduces FN and the FN/(FN + FP) ratio across all models, effectively mitigating false negatives caused by class imbalance. More importantly, ATSL consistently improves AUC, with an average gain of 4.53, and achieves the largest increases in BERT$_{base}$ TTM-RE (from 68.05 to 73.55) and RoBERTa$_{large}$ ATLOP (from 65.61 to 72.50). These improvements in AUC suggest that ATSL not only reduces false negatives but also improves overall discriminative ability and robustness under class imbalance. We also observe that FP increases in some models, but the overall performance gain remains substantial.

Table 8: Experimental results on class imbalance on Re-DocRED dev set. FN (False Negative): Predicts a positive example as negative. FP (False Positive): Predicts a negative example as positive. FN_NA: Predicts a positive example as negative, and the predicted label is NA.

| Model | FN↓ | FP↓ | FN_NA | FN / (FN + FP) | AUC↑ | Model | FN↓ | FP↓ | FN_NA | FN / (FN + FP) | AUC↑ |
|---|---|---|---|---|---|---|---|---|---|---|---|
| BERT$_{base}$ Encoder | | | | | | RoBERTa$_{large}$ Encoder | | | | | |
| ATLOP Zhou et al. (2021) | 5833 | 1942 | 5054 | 0.75 | 60.81 | ATLOP Zhou et al. (2021) | 5205 | 1741 | 4405 | 0.75 | 65.61 |
| ATLOP + ATSL (ours) | 4181 | 4029 | 3467 | 0.51 | **67.66**(6.85↑) | ATLOP + ATSL (ours) | 3695 | 2951 | 3052 | 0.56 | **72.50**(6.89↑) |
| DocuNet Zhang et al. (2021) | 5777 | 1955 | 4971 | 0.75 | 63.90 | DocuNet Zhang et al. (2021) | 5126 | 1833 | 4203 | 0.74 | 67.73 |
| DocuNet + ATSL (ours) | 4111 | 4154 | 3428 | 0.50 | **68.04**(4.14↑) | DocuNet + ATSL (ours) | 4069 | 2592 | 3362 | 0.61 | **70.32**(2.59↑) |
| TTM-RE Gao et al. (2024) | 4413 | 3745 | 3724 | 0.54 | 68.05 | TTM-RE Gao et al. (2024) | 4245 | 2526 | 3504 | 0.63 | 70.04 |
| TTM-RE + ATSL (ours) | 3710 | 3107 | 2898 | 0.54 | **73.55**(5.50↑) | TTM-RE + ATSL (ours) | 3766 | 2156 | 3020 | 0.64 | **74.09**(4.05↑) |
| VaeDiff-DocRE Tran et al. (2025) | 5180 | 2525 | 4358 | 0.67 | 66.28 | VaeDiff-DocRE Tran et al. (2025) | 4974 | 1874 | 4099 | 0.73 | 68.12 |
| VaeDiff-DocRE + ATSL (ours) | 4313 | 3354 | 3664 | 0.56 | **69.18**(2.90↑) | VaeDiff-DocRE + ATSL (ours) | 4146 | 2636 | 3467 | 0.61 | **71.46**(3.34↑) |

## 6.3 RESOURCE EFFICIENCY

To assess the resource efficiency of MD-RE, **Table 9** presents a comparison of memory usage and training time. MD-RE demonstrates favorable efficiency, requiring only 17.63 GiB of memory, which is lower than that of TTM-RE (without evidence), as well as Eider and SAIS (both of which use evidence). Its training time of 81.80 minutes is also significantly shorter than that of KD-DocRE and TTM-RE, both of which do not use evidence.

Table 9: Resource usage under BERT$_{base}$ on Re-DocRED (batch size 4). * from Ma et al. (2023).

| Method | Memory (GiB) | Training Time (Min) | F1 |
|---|---|---|---|
| without evidence | | | |
| ATLOP | 10.37 | 55.80 | 74.02 |
| KD-DocRE | 14.71 | 169.59 | 74.55 |
| TTM-RE | 20.30 | 208.37 | - |
| with evidence | | | |
| Eider | 43.10* | - | - |
| SAIS | 46.20* | - | - |
| DREEAM | 13.46 | 59.02 | 74.23 |
| MD-RE (ours) | 17.63 | 81.80 | **77.80** |

Table 10: Training time comparison of ATSL and other losses on the ATLOP backbone with BERT$_{base}$, trained for 30 epochs on Re-DocRED (batch size 4).

| Loss Function | Approx. Training Time |
|---|---|
| ATL Zhou et al. (2021) | 59.71 minutes |
| Balanced-Softmax Zhang et al. (2021) | 57.27 minutes |
| AML Wei & Li (2022) | 54.50 minutes |
| AFL Tan et al. (2022a) | 60.45 minutes |
| HingeABL$_{SAT}$ Wang et al. (2023) | 58.39 minutes |
| HingeABL$_{MeanSAT}$ Wang et al. (2023) | 60.81 minutes |
| HingeABL Wang et al. (2023) | 59.87 minutes |
| ATSL (Ours) | 57.12 minutes |

## 6.4 TRAINING COST OF ATSL LOSS

To evaluate the training cost of the proposed ATSL, we compare the training time of ATSL and several other losses. As shown in **Table 10**, ATSL requires 57.12 minutes, which is largely comparable to other existing losses. For example, ATL and HingeABL require 59.71 and 59.87 minutes, respectively, while Balanced-Softmax and AML complete training in 57.27 and 54.50 minutes. These results indicate that incorporating ATSL does not introduce any notable computational overhead.

## 7 CONCLUSION

We propose a novel MD-RE framework that incorporates three discriminators with different decision criteria. By leveraging unique characteristics of each discriminator, we integrate their outputs using a weighted fusion method. This design alleviates the issue of introducing noise unrelated to the relation from the entire document and does not rely on evidence sentences. In addition, we propose a novel multi-label classification loss, ATSL, which effectively mitigates class imbalance. Notably, MD-RE and ATSL are independent yet complementary, each capable of improving DocRE on its own without interdependence. Extensive experiments show the effectiveness and general applicability of both MD-RE framework and ATSL loss.

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

# A    THEORETICAL ANALYSIS OF ATSL LOSS

In this section, we provide a rigorous theoretical justification for the **A**daptive **T**hreshold **S**hifted **L**oss (**ATSL**), including convexity, conditional risk analysis, optimal logits, and Bayes consistency.

## A.1    PROBLEM SETUP AND NOTATION

Let $\mathcal{X}$ be the input space and $\mathcal{Y} = \{0, 1\}^K$ be the multi-label output space, where $K$ is the number of labels. For an input $x \in \mathcal{X}$, let $y \in \mathcal{Y}$ denote its corresponding multi-label vector, where $y_r = 1$ if label $r$ is relevant and $y_r = 0$ otherwise.

Define $\Delta_r(x) = \mathbb{P}(y_r = 1 \mid x)$ as the conditional probability that label $r$ is relevant given input $x$. We assume conditional independence of labels given $x$, i.e., $\mathbb{P}(y \mid x) = \prod_{r=1}^{K} \Delta_r(x)^{y_r}(1 - \Delta_r(x))^{1-y_r}$.

Given a prediction function $f : \mathcal{X} \to \mathbb{R}^{K+1}$ that outputs logits $(\text{logit}_1, \ldots, \text{logit}_K, \text{logit}_{\text{TH}})$. Let $y \in \{0, 1\}^K$ be the ground-truth label vector for input $x$. We define the sets of positive and negative classes for loss calculation as:

$$\mathcal{P}_T = \{r : y_r = 1\} \quad \text{(ground-truth positive labels)}, \tag{17}$$
$$\mathcal{N}_T = \{r : y_r = 0\} \quad \text{(ground-truth negative labels)}. \tag{18}$$

## A.2    ATSL LOSS DEFINITION

The ATSL loss is defined as:

$$\mathcal{L}_1' = -\sum_{r \in \mathcal{P}_T} \log \frac{\exp(\text{logit}_r)}{\exp(\text{logit}_{\text{TH}} + \beta) + \sum_{r' \in \mathcal{P}_T} \exp(\text{logit}_{r'})}, \tag{19}$$

$$\mathcal{L}_2' = -\log \frac{\exp(\text{logit}_{\text{TH}} + \lambda)}{\exp(\text{logit}_{\text{TH}} + \lambda) + \sum_{r' \in \mathcal{N}_T} \exp(\text{logit}_{r'})}, \tag{20}$$

$$\mathcal{L}_{\text{ATSL}} = \mathcal{L}_1' + \mathcal{L}_2', \tag{21}$$

where $\mathcal{P}_T$ and $\mathcal{N}_T$ are the ground-truth positive and negative sets as defined in **Section 3.1**, and $\beta, \lambda \geq 0$ are hyperparameters.

## A.3 Convexity of ATSL

**Proposition 1 (Convexity).** The ATSL loss can be rewritten as:

$$\mathcal{L}_1' = \sum_{r \in \mathcal{P}_T} \left[ \log \left( \exp(\text{logit}_{\text{TH}} + \beta) + \sum_{r' \in \mathcal{P}_T} \exp(\text{logit}_{r'}) \right) - \text{logit}_r \right], \tag{22}$$

$$\mathcal{L}_2' = \log \left( \exp(\text{logit}_{\text{TH}} + \lambda) + \sum_{r' \in \mathcal{N}_T} \exp(\text{logit}_{r'}) \right) - (\text{logit}_{\text{TH}} + \lambda). \tag{23}$$

**Proof:** The function log-sum-exp$(z_1, \ldots, z_n) = \log(\sum_{i=1}^n \exp(z_i))$ is convex Boyd & Vandenberghe (2004) in $(z_1, \ldots, z_n)$. Since $\mathcal{L}_1'$ and $\mathcal{L}_2'$ are compositions of convex functions (log-sum-exp) and affine transformations of the logits, both are convex functions over the domain $\mathbb{R}^{K+1}$. Therefore, $\mathcal{L}_{\text{ATSL}} = \mathcal{L}_1' + \mathcal{L}_2'$ is convex.

## A.4 Conditional Risk Analysis

The conditional risk associated with the ATSL loss is:

$$\mathcal{R}_{\text{ATSL}}(f \mid x) = \mathbb{E}[\mathcal{L}_{\text{ATSL}}(f(x), Y) \mid X = x]. \tag{24}$$

**Proposition 2 (Conditional Risk Formula).** Under the conditional independence assumption, the conditional risk can be expressed as:

$$\mathcal{R}_{\text{ATSL}}(f \mid x) = \mathbb{E} \left[ \sum_{r \in \mathcal{P}_T} \log \left( \exp(\text{logit}_{\text{TH}} + \beta) + \sum_{r' \in \mathcal{P}_T} \exp(\text{logit}_{r'}) \right) - \sum_{r \in \mathcal{P}_T} \text{logit}_r \right]$$
$$+ \mathbb{E} \left[ \log \left( \exp(\text{logit}_{\text{TH}} + \lambda) + \sum_{r' \in \mathcal{N}_T} \exp(\text{logit}_{r'}) \right) - (\text{logit}_{\text{TH}} + \lambda) \right]. \tag{25}$$

For a given input $x$ and its deterministic ground-truth labels, the expectation simplifies to the point-wise loss:

$$\mathcal{R}_{\text{ATSL}}(f \mid x) = |\mathcal{P}_T| \log \left( \exp(\text{logit}_{\text{TH}} + \beta) + \sum_{r' \in \mathcal{P}_T} \exp(\text{logit}_{r'}) \right) - \sum_{r \in \mathcal{P}_T} \text{logit}_r$$
$$+ \log \left( \exp(\text{logit}_{\text{TH}} + \lambda) + \sum_{r' \in \mathcal{N}_T} \exp(\text{logit}_{r'}) \right) - (\text{logit}_{\text{TH}} + \lambda), \tag{26}$$

where $|\mathcal{P}_T|$ denotes the cardinality of $\mathcal{P}_T$.

## A.5 Optimal Logits

**Proposition 3 (Optimal Logits).** The population-optimal logits that minimize $\mathcal{R}_{\text{ATSL}}(f \mid x)$ satisfy the following first-order conditions:

$$\frac{\partial \mathcal{R}_{\text{ATSL}}}{\partial \text{logit}_r} = |\mathcal{P}_T| \cdot \frac{\exp(\text{logit}_r)}{\exp(\text{logit}_{\text{TH}} + \beta) + \sum_{r' \in \mathcal{P}_T} \exp(\text{logit}_{r'})} - 1 = 0, \qquad r \in \mathcal{P}_T,$$

$$\frac{\partial \mathcal{R}_{\text{ATSL}}}{\partial \text{logit}_s} = \frac{\exp(\text{logit}_s)}{\exp(\text{logit}_{\text{TH}} + \lambda) + \sum_{r' \in \mathcal{N}_T} \exp(\text{logit}_{r'})} = 0, \qquad s \in \mathcal{N}_T,$$

$$\frac{\partial \mathcal{R}_{\text{ATSL}}}{\partial \text{logit}_{\text{TH}}} = |\mathcal{P}_T| \cdot \frac{\exp(\text{logit}_{\text{TH}} + \beta)}{\exp(\text{logit}_{\text{TH}} + \beta) + \sum_{r' \in \mathcal{P}_T} \exp(\text{logit}_{r'})}$$
$$+ \frac{\exp(\text{logit}_{\text{TH}} + \lambda)}{\exp(\text{logit}_{\text{TH}} + \lambda) + \sum_{r' \in \mathcal{N}_T} \exp(\text{logit}_{r'})} - 1 = 0.$$

**Note:** The second condition implies $\text{logit}_s \to -\infty$ for $s \in \mathcal{N}_T$, which aligns with the intuition that labels predicted as negative should have very low logits. The first and third conditions provide relationships between the positive label logits and the threshold.

### A.6 BAYES CONSISTENCY

**Proposition 4 (Bayes Consistency).** ATSL is Bayes consistent in the sense that it provides a consistent ranking-based approach for multi-label classification.

**Proof:**

*Step 1: Convergence of Empirical Risk Minimizer.* Since ATSL is convex and differentiable, by standard results in statistical learning theory (Vapnik, 1999; Shalev-Shwartz & Ben-David, 2014), the empirical risk minimizer converges almost surely to the population risk minimizer as the sample size approaches infinity.

*Step 2: Optimal Threshold Property.* The ATSL loss encourages:

- Labels in $\mathcal{P}_T$ to have high logits relative to the threshold $\text{logit}_{\text{TH}} + \beta$
- The threshold $\text{logit}_{\text{TH}} + \lambda$ to be high relative to labels in $\mathcal{N}_T$

At the population optimum, this creates a clear separation between positive and negative predictions, with the threshold adapting to provide optimal ranking performance.

*Step 3: Ranking-Induced Consistency.* Unlike margin-based pairwise losses that assume balanced label distributions, the ATSL loss introduces tunable parameters $\beta$ and $\lambda$ to handle label imbalance explicitly. These parameters shift the decision threshold to reflect the asymmetry between positive and negative labels.

The ATSL loss encourages positive labels ($\mathcal{P}_T$) to be ranked above both the threshold and the negative labels ($\mathcal{N}_T$), creating a separation margin that adapts to the imbalance.

Assuming the optimal ranking is induced by the ordering of the conditional label probabilities, the ATSL loss induces a scoring function that is consistent with this ordering. Hence, the classifier converges to the Bayes-optimal ranking under a label-ranking loss, even when $\beta, \lambda > 0$ are fixed and used to regularize the imbalance.

## B DATASETS AND HYPERPARAMETERS

### B.1 HYPERPARAMETER SETTINGS

We summarize the hyperparameters used for training on three datasets in **Table 11**. To ensure robust evaluation, we conduct experiments using three different random seeds and report the averaged results. The batch size is set to 4, and learning rates for the encoder and classifier are tuned separately: a lower rate (1e-5 to 5e-5) for the encoder and a higher rate (1e-4) for the classifier. A warm-up ratio of 0.06 is applied across all settings to stabilize the early stages of training. The ATSL-related hyperparameters include $\boldsymbol{\lambda} = \{\lambda_{\text{Recall}}, \lambda_{\text{Coarse}}, \lambda_{\text{Fine}}\}$ and $\boldsymbol{\beta} = \{\beta_{\text{Recall}}, \beta_{\text{Coarse}}, \beta_{\text{Fine}}\}$, which control the weights for the recall, coarse, and fine discriminators, respectively. Since the hyperparameters $\lambda$ and $\beta$ in the ATSL loss serve the same purpose, we adjust only $\lambda$ and set $\beta$ to 0 to streamline the process. The number of training epochs is determined based on the dataset, ranging from 8 epochs on DocRED Yao et al. (2019) to 30 epochs on DWIE Zaporojets et al. (2021).

### B.2 DATASETS

The statistics of the three datasets are shown in **Table 12**, with detailed descriptions provided below.

**DocRED** Yao et al. (2019) is a large-scale, human-annotated dataset constructed from Wikipedia, specifically designed for DocRE task. Although it contains 3,053 documents for training and 1,000 documents each for development and testing, it suffers from a substantial number of missing annotations. To address this issue, we adopt the Re-DocRED Tan et al. (2022b) dataset for evaluation on the dev and test sets.

**DWIE** Zaporojets et al. (2021) is a multi-task dataset focused on entity-centric tasks, with 602 documents in the train set, 98 in the dev set, and 99 in the test set.

**Re-DocRED** Tan et al. (2022b) is a revised version of the DocRED Yao et al. (2019) dataset, which has been reprocessed and manually validated to address the numerous false negative issues present

Table 11: Hyperparameters.

| Dataset | Re-DocRED | | DocRED | DWIE |
|---|---|---|---|---|
| | **BERT** | **RoBERTa** | **RoBERTa** | **BERT** |
| epoch | 20 | 20 | 8 | 30 |
| lr_encoder | 4e-5 | 2e-5 | 1e-5 | 5e-5 |
| lr_classifier | 1e-4 | 1e-4 | 1e-4 | 1e-4 |
| batch size | 4 | 4 | 4 | 4 |
| warmup_ratio | 0.06 | 0.06 | 0.06 | 0.06 |
| $\rho$ | 4 | 4 | 4 | 4 |
| $\alpha$ | 0.65 | 0.60 | 1.0 | 1.10 |
| $\lambda_{\text{Recall}}$ | 3.0 | 3.0 | 4.5 | 3.0 |
| $\lambda_{\text{Coarse}}$ | 1.5 | 1.5 | 4.0 | 1.5 |
| $\lambda_{\text{Fine}}$ | 0.0 | 0.0 | - | 0.0 |
| $\beta_{\text{Recall}}$ | 0.0 | 0.0 | 0.0 | 0.0 |
| $\beta_{\text{Coarse}}$ | 0.0 | 0.0 | 0.0 | 0.0 |
| $\beta_{\text{Fine}}$ | 0.0 | 0.0 | - | 0.0 |

in the original DocRED. Additionally, the validation and test sets of Re-DocRED are derived by splitting the dev set of DocRED, with each containing 500 documents. The number of documents in the train set of Re-DocRED remains the same as in DocRED, with a total of 3,053 documents.

Table 12: Statistics of datasets.

| Dataset | Split | #Docs. | #Entities. | #Rels. | Long Docs ($>$10 sents) |
|---|---|---|---|---|---|
| DocRED | train | 3,053 | 59,493 | 96 | 499 |
| | dev[†] | 500 | 9,684 | 96 | 87 |
| | test[†] | 500 | 9,779 | 96 | 78 |
| DWIE | train | 602 | 16,494 | 65 | 553 |
| | dev | 98 | 2,785 | 65 | 87 |
| | test | 99 | 2,623 | 65 | 92 |
| Re-DocRED | train | 3,053 | 59,359 | 96 | 499 |
| | dev[†] | 500 | 9,684 | 96 | 87 |
| | test[†] | 500 | 9,779 | 96 | 78 |

## C  FURTHER ABLATION STUDY

**Effect of Discriminator.**    To investigate the contributions of each discriminator in our MD-RE framework, we conduct ablation experiments on Re-DocRED and DWIE. **Table 13** summarizes the results. With only the recall discriminator, most results are inferior to those obtained using two or three discriminators. *These results further demonstrate the advantages of our discriminator design.* The recall discriminator alone is not sufficient to effectively handle more difficult negative or ambiguous samples. In contrast, the coarse and fine discriminators apply progressively stricter, more refined criteria, enabling them to better handle these challenging samples. Notably, the complementary filtering effect of the three discriminators leads to the best performance.

**Sample Selection Strategy.**    To evaluate the effect of LNS, we compare it with random sampling and margin-based mining. As shown in **Table 14**, LNS consistently achieves the best F1 and Ign-F1 scores. These results indicate that, although the gains are modest, LNS can also work in conjunction with the ATSL loss to dynamically adjust the threshold and recall of each discriminator.

Table 13: Ablation study of *discriminators* on the Re-DocRED and DWIE dev sets.

| Model | F1 | Ign-F1 |
|---|---|---|
| MD-RE on Re-DocRED using BERT$_{base}$ (ours) | **77.70** | **76.46** |
| *w/o* Coarse Discriminator | 77.18 | 75.67 |
| *w/o* Fine Discriminator | 76.38 | 74.58 |
| *w/o* Coarse & Fine Discriminator (*only* Recall Discriminator) | 76.23 | 74.83 |
| MD-RE on Re-DocRED using RoBERTa$_{large}$ (ours) | **81.44** | **80.38** |
| *w/o* Coarse Discriminator | 80.70 | 79.33 |
| *w/o* Fine Discriminator | 80.79 | 79.39 |
| *w/o* Coarse & Fine Discriminator (*only* Recall Discriminator) | 80.35 | 79.22 |
| MD-RE on DWIE using BERT$_{base}$ (ours) | **73.81** | **68.37** |
| *w/o* Coarse Discriminator | 72.67 | 68.24 |
| *w/o* Fine Discriminator | 72.39 | 67.70 |
| *w/o* Coarse & Fine Discriminator (*only* Recall Discriminator) | 69.14 | 61.11 |
| MD-RE on DWIE using RoBERTa$_{large}$ (ours) | **77.11** | **73.13** |
| *w/o* Coarse Discriminator | 76.37 | 72.35 |
| *w/o* Fine Discriminator | 76.67 | 72.47 |
| *w/o* Coarse & Fine Discriminator (*only* Recall Discriminator) | 76.18 | 70.64 |

Table 14: Ablation study of *sample selection strategies* on the Re-DocRED dev set using BERT$_{base}$ as the encoder.

| Model | F1 | Ign-F1 |
|---|---|---|
| MD-RE (ours) | | |
| *w* LNS | **77.70** | **76.46** |
| *w* Random sampling | 77.57 | 76.39 |
| *w* Margin-based mining | 77.44 | 75.87 |

**Fusion Strategy.** To evaluate the effect of Weighted Fusion, we compare it with several alternative fusion strategies, including Pipeline Fusion, Add Fusion, Mean Fusion, and Voting Fusion. As shown in **Table 15**, Weighted Fusion yielded the most favorable performance, achieving the highest F1 score of 77.70 and Ign-F1 score of 76.46. This represents a notable advantage, surpassing the runner-up Add Fusion by 0.68 F1 points. These results indicate that Weighted Fusion more effectively integrates information than simple addition, averaging, or voting, leading to improved overall performance.

Table 15: Ablation study of *fusion strategies* on the Re-DocRED dev set using BERT$_{base}$ as the encoder, where *w* indicates inclusion.

| Model | F1 | Ign-F1 |
|---|---|---|
| MD-RE (ours) | | |
| *w* Weighted Fusion | **77.70** | **76.46** |
| *w* Pipeline Fusion | 72.83 | 72.27 |
| *w* Add Fusion | 77.02 | 75.53 |
| *w* Mean Fusion | 76.05 | 75.21 |
| *w* Voting Fusion | 72.63 | 72.08 |

# D  FURTHER ANALYSIS

Table 16: Results of the MD-RE evaluated on Re-DocRED documents longer than 10 sentences.

| Model | dev F1 | test F1 | Model | dev F1 | test F1 |
|---|---|---|---|---|---|
| BERT$_{base}$ Encoder | | | RoBERTa$_{large}$ Encoder | | |
| ATLOP Zhou et al. (2021) | 70.44 | 73.67 | ATLOP Zhou et al. (2021) | 73.92 | 78.36 |
| DocuNet Zhang et al. (2021) | 70.17 | 71.81 | DocuNet Zhang et al. (2021) | 75.01 | 76.54 |
| TTM-RE Gao et al. (2024) | 71.93 | 73.01 | TTM-RE Gao et al. (2024) | 77.52 | 79.96 |
| VaeDiff-DocRE Tran et al. (2025) | 70.27 | 73.38 | VaeDiff-DocRE Tran et al. (2025) | 74.47 | 77.09 |
| MD-RE (ours) | **73.54** | **77.32** | MD-RE (ours) | **78.07** | **80.76** |

## D.1 Performance of MD-RE framework on Long Documents

To evaluate the capability of MD-RE with ATSL on long documents (more than 10 sentences), we evaluate it on the Re-DocRED dataset and compare it against strong baselines. **Table 16** shows that MD-RE consistently outperforms all baselines on long documents. Using the BERT$_{base}$ encoder, our model achieves 73.54 F1 on dev and 77.32 F1 on test, outperforming the strongest baseline (TTM-RE) by 1.61 and 4.31 points. Using the RoBERTa$_{large}$ encoder, MD-RE reaches 78.07 F1 on dev and 80.76 F1 on test, again showing substantial improvements over previous methods. These results indicate that MD-RE with ATSL is particularly effective at capturing and aggregating information across long text spans, leading to improved relation extraction performance on long documents.

## D.2 Impact of Discriminator Number on MD-RE Framework

To evaluate the impact of the number of discriminators on the MD-RE framework, we conduct experiments using different discriminator counts on the Re-DocRED test set with BERT$_{base}$. As shown in **Fig. 3**, the F1 score improves as the number of discriminators increases from 1 to 3, reaching the highest performance at 3 discriminators (77.80). However, further increasing the number to 4 and 5 leads to a decline in performance, with a sharp drop observed at 5 discriminators (73.71). These results suggest that a moderate number of discriminators is beneficial, while too many may introduce redundancy or overfitting, degrading the model's effectiveness.

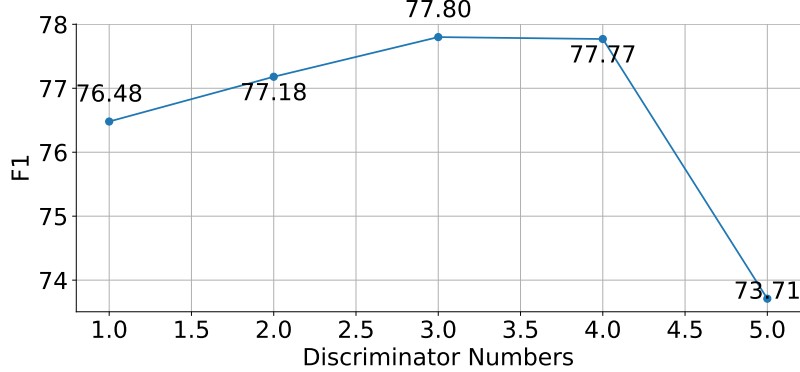

Figure 3: Performance comparison of different numbers of discriminators in the MD-RE framework on the Re-DocRED test set using BERT$_{base}$.

## D.3 Effect of Hyperparameter $\lambda$ in ATSL

To evaluate the impact of the hyperparameter $\lambda$ in ATSL, we vary its value from 0.0 to 4.0 and systematically analyze its effect on performance. As shown in **Fig. 4**, the F1 consistently improves with increasing $\lambda$, achieving the highest value of 76.23 at $\lambda = 3.0$. A marginal decline in performance is observed beyond this point. These observations suggest that while ATSL is generally robust to the choice of $\lambda$, careful tuning around $[2.0, 3.5]$ is beneficial for maximizing performance.

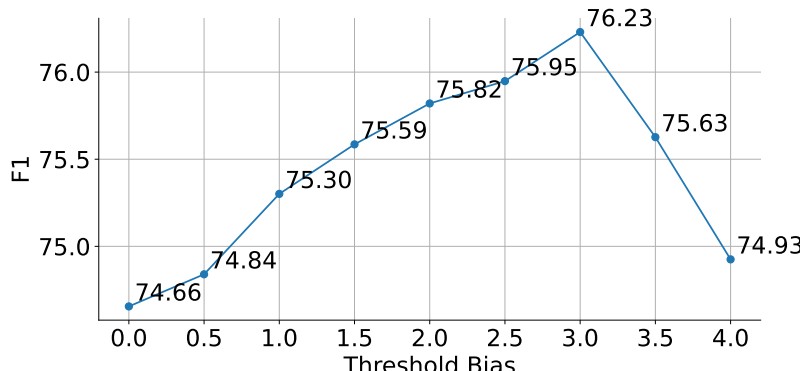

Figure 4: Performance under different $\lambda$ values of the ATSL loss on the Re-DocRED dev set, using ATLOP and BERT$_{base}$.

### D.4 EFFECT OF HYPERPARAMETER $\alpha$ IN MD-RE WITH ATSL LOSS

To further investigate the effect of different $\alpha$ values on MD-RE performance, we conduct additional experiments on the Re-DocRED and DWIE datasets, as shown in **Fig. 5**. On the Re-DocRED dev set, the F1 score gradually increases as $\alpha$ rises from 0.5 to 0.65, reaching a peak of 77.70. Beyond this point, further increases in $\alpha$ lead to a decline in performance. On the DWIE dev set, the F1 score peaks at $\alpha = 1.1$ with a value of 73.81, showing a similar trend of first rising and then falling. These results indicate that the optimal value of $\alpha$ varies across datasets but is generally tuned around 1.0. This setting ensures effective model fusion while reasonably balancing the contribution of each discriminator.

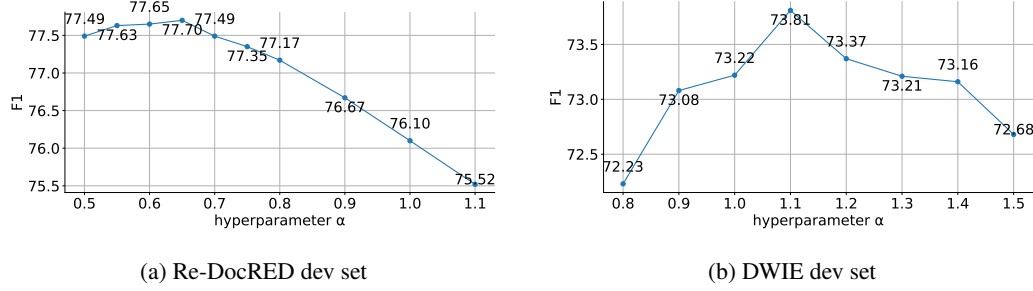

(a) Re-DocRED dev set          (b) DWIE dev set

Figure 5: Performance under different $\alpha$ values of the MD-RE with ATSL loss, using BERT$_{base}$. Results are shown for both the Re-DocRED and DWIE dev sets.

### D.5 CASE STUDY AND ERROR ANALYSIS

#### D.5.1 COMPREHENSIVE CASE STUDY

To better illustrate the effectiveness and interpretability of our MD-RE framework, we present a representative example in **Fig. 6**, where we provide a detailed comparison between the results of our MD-RE framework and those of the baseline model ATLOP. In the predictions of the ATLOP model, only two triples were correctly identified, with four triples missing. In the predictions from the recall discriminator combined with the coarse discriminator, compared to ATLOP, two additional triples were correctly predicted. However, the triples (**Royal Navy**, P607, **World War II**) and (**World War II**, P710, **Royal Navy**) were still missed. Moreover, the incorrect predictions (**Britain**, P607, **World War II**) and (**World War II**, P156, **World War I**) were introduced. This is because the recall discriminator focuses on improving recall, leading to some triples being over-recalled, while the coarse discriminator emphasizes initial filtering and reducing excessive predictions, which

introduces incorrect predictions. In the predictions from the recall discriminator combined with the fine discriminator, the previously missed triples (**Royal Navy**, P607, **World War II**) and (**World War II**, P710, **Royal Navy**) were successfully recovered, and the incorrect prediction (**World War II**, P156, **World War I**) was corrected. This indicates that the fine discriminator is more effective at filtering out irrelevant or incorrect triples, thus improving prediction accuracy. Finally, by integrating the recall, coarse, and fine discriminators, the full MD-RE framework successfully predicted all the triples. Furthermore, the incorrect prediction (**Britain**, P607, **World War II**) from the combined recall and fine discriminator was eliminated. These results demonstrate that the integration of all three discriminators substantially enhances the overall performance of the framework.

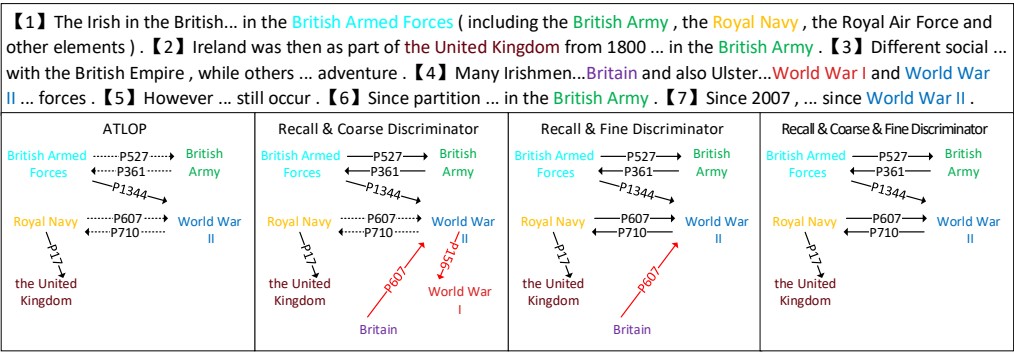

Figure 6: Case study of the baseline model ATLOP and our proposed MD-RE framework on Re-DocRED dev set. Black solid lines indicate correct predictions, red solid lines represent incorrect predictions, and dashed lines denote missing predictions.

### D.5.2 ERROR ANALYSIS OF MD-RE

To further investigate the failure patterns of MD-RE, we analyze model performance across different relation types. As shown in **Table 17**, MD-RE exhibits strong performance on high-frequency relations such as *located in the administrative territorial entity* and *country*, achieving F1 above 80. However, performance drops substantially on long-tail relations, including *replaced by* (22.22) and *replaces* (12.50), indicating that most performance degradation originates from relations with limited training data. In addition, we observe that false positive errors occur more often in high-frequency relations, indicating occasional over-prediction.

Furthermore, to more thoroughly evaluate the effectiveness of our MD-RE framework in handling long-tail relations, we conduct comparative experiments on the Re-DocRED dataset and report performance separately on frequent (Freq) and long-tail (LTail) relation subsets. As shown in **Table 18**, our method consistently improves the F1 score for long-tail relations across different models. Specifically, MD-RE improves LTail F1 by 2.50 on the dev set and 3.02 on the test set with BERT$_{base}$, and by 2.67 and 2.60 with RoBERTa$_{large}$. These results demonstrate the robustness and generalization ability of MD-RE in modeling sparse relations, while simultaneously achieving overall performance gains without degrading frequent-relation performance.

### D.5.3 ERROR ANALYSIS OF FALSE POSITIVES AFTER APPLYING ATSL

To investigate which types of false positives increase after applying ATSL, we examine false positives (FPs) by relation type on the Re-DocRED dev set. As shown in **Table 19**, the results show that FP increases mainly occur in high-frequency relations, such as *located in the administrative territorial entity* and *country*, whereas long-tail relations exhibit smaller growth. This suggests that ATSL tends to over-predict frequent relations, improving recall at the cost of higher FPs.

Table 17: Head vs. Tail relation performance of our MD-RE framework on the Re-DocRED dev set with BERT$_{base}$.

| Type | Relation Name | FP | FN | Total | F1 | Rank (by train set) |
|------|---------------|----|----|-------|-----|---------------------|
| **Head** | located in the administrative territorial entity | 635 | 696 | 3846 | 82.56 | 1 |
| | country | 414 | 449 | 2739 | 84.14 | 2 |
| | country of citizenship | 194 | 147 | 823 | 79.86 | 3 |
| | contains administrative territorial entity | 114 | 155 | 662 | 79.03 | 4 |
| | notable work | 82 | 127 | 604 | 82.03 | 5 |
| | has part | 75 | 196 | 469 | 66.83 | 6 |
| | part of | 77 | 192 | 444 | 65.20 | 7 |
| | *... (intermediate relations omitted) ...* | | | | | |
| **Tail** | manufacturer | 15 | 20 | 76 | 76.19 | 80 |
| | parent taxon | 3 | 5 | 28 | 85.19 | 81 |
| | original language of work | 3 | 9 | 19 | 62.50 | 82 |
| | end time | 10 | 16 | 26 | 43.48 | 83 |
| | location of formation | 1 | 9 | 24 | 75.00 | 84 |
| | replaced by | 3 | 11 | 13 | 22.22 | 85 |
| | replaces | 2 | 12 | 13 | 12.50 | 86 |
| | narrative location | 2 | 1 | 7 | 80.00 | 87 |

Table 18: Long-Tail relation performance with different models. Experimental result on Re-DocRED dataset. * from Tran et al. (2025), and † our reproduced results. **Freq F1** evaluates the 10 most common relations, while **LTail F1** covers the remaining 86 uncommon relations in Re-DocRED.

| Model | Dev | | | | Test | | | |
|-------|-----|-----|---------|----------|------|-----|---------|----------|
| | F1 | Ign-F1 | Freq F1 | LTail F1 | F1 | Ign-F1 | Freq F1 | LTail F1 |
| *BERT$_{base}$ Encoder* | | | | | | | | |
| ATLOP Zhou et al. (2021) | 74.22 | 73.35 | 77.77† | 68.83† | 74.02 | 73.22 | 75.92* | 67.46* |
| DocuNET Zhang et al. (2021) | 74.65 | 73.68 | 78.55† | 69.41† | 74.49 | 73.60 | 77.54† | 68.33† |
| KD-DocRE Tan et al. (2022a) | 74.69 | 73.76 | 78.71* | 70.26* | 74.55 | 73.67 | 78.28* | 69.52* |
| TTM-RE Gao et al. (2024) | 76.21 | 74.74 | 80.45† | 69.42† | 76.33 | 74.89 | 79.83† | 68.99† |
| VaeDiff-DocRE Tran et al. (2025) | 75.89* | 74.96* | 79.18* | 71.12* | 75.07* | 74.13* | 78.32* | 70.06* |
| MD-RE (ours) | **77.70** (1.49↑) | **76.46** (1.50↑) | **80.51** (0.06↑) | **73.62** (2.50↑) | **77.80** (1.47↑) | **76.63** (1.74↑) | **80.81** (0.98↑) | **73.18** (3.02↑) |
| *RoBERTa$_{large}$ Encoder* | | | | | | | | |
| ATLOP Zhou et al. (2021) | 77.63 | 76.88 | 80.92† | 72.83† | 77.73 | 76.94 | 80.78* | 72.29* |
| DocuNET Zhang et al. (2021) | 78.16 | 77.53 | 80.73† | 73.34† | 77.92 | 77.27 | 81.25* | 73.32* |
| KD-DocRE Tan et al. (2022a) | 78.65 | 77.92 | 80.52† | 72.54† | 78.35 | 77.63 | 80.85* | 74.31* |
| TTM-RE Gao et al. (2024) | 78.13 | 78.05 | 83.05† | 73.90† | 79.95 | 78.20 | 83.43† | 73.23† |
| VaeDiff-DocRE Tran et al. (2025) | 79.19* | 78.35* | 82.00* | 75.13* | 79.03* | 78.22* | 81.84* | 74.73* |
| MD-RE (ours) | **81.44** (2.25↑) | **80.38** (2.03↑) | **84.10** (1.05↑) | **77.80** (2.67↑) | **81.49** (1.54↑) | **80.45** (2.23↑) | **84.10** (0.67↑) | **77.33** (2.60↑) |

Table 19: Analysis of increased false positives by relation type after applying ATSL on Re-DocRED dev set.

| Relation Name | ATLOP (FP) | ATLOP with ATSL (FP) | $\Delta$FP | Rank (by train set) |
|---|---|---|---|---|
| **BERT$_{base}$ Encoder** | | | | |
| located in the administrative territorial entity | 411 | 716 | 305 | 1 |
| country | 283 | 449 | 166 | 2 |
| country of citizenship | 135 | 222 | 87 | 3 |
| contains administrative territorial entity | 57 | 143 | 86 | 4 |
| part of | 40 | 113 | 73 | 7 |
| applies to jurisdiction | 28 | 79 | 51 | 11 |
| has part | 53 | 104 | 51 | 6 |
| publication date | 38 | 85 | 47 | 9 |
| member of | 32 | 78 | 46 | 10 |
| owned by | 11 | 53 | 42 | 38 |
| **RoBERTa$_{large}$ Encoder** | | | | |
| located in the administrative territorial entity | 355 | 490 | 135 | 1 |
| country | 255 | 346 | 91 | 2 |
| part of | 49 | 106 | 57 | 7 |
| continent | 23 | 63 | 40 | 20 |
| has part | 54 | 93 | 39 | 6 |
| contains administrative territorial entity | 53 | 91 | 38 | 4 |
| member of | 23 | 60 | 37 | 10 |
| country of citizenship | 124 | 158 | 34 | 3 |
| headquarters location | 6 | 37 | 31 | 55 |
| country of origin | 33 | 63 | 30 | 16 |

## E  USE OF LARGE LANGUAGE MODELS (LLMS)

Large language models are used solely for language polishing and grammar correction of the paper draft. All technical content, including the research idea, methodology, experiments, and writing, is solely produced by the authors.

