# OpenReview forum: "MD-RE: A Multi-Discrimination Framework for Document-Level Relation Extraction with Adaptive Threshold Shifted Loss"
_ICLR.cc/2026/Conference — Submitted to ICLR 2026_

### Official Review · Reviewer_9XJh · 2025-10-16

**Soundness:** 3
**Presentation:** 3
**Contribution:** 2
**Rating:** 4
**Confidence:** 4

**Summary:**

This paper points out that for the document-level relation extraction task, existing methods face challenges in identifying the relationships between entities. This paper proposes a Multi-Discrimination framework (MD-RE) for DocRE and incorporates a perceptual loss function to improve the model's performance on the DocRE task.

**Strengths:**

1. This paper is well-written, with clear logic, and it fully conveys the motivation and methodology of the research.

2. The paper conducts extensive experiments, including comparative experiments on models of different types and scales.

**Weaknesses:**

1. First, the paper points out that MD-RE will improve the model's performance on the DocRE task. However, MD-RE adopts multiple discriminators, and the additional computational and time costs incurred have not been evaluated. This makes comparisons with previous methods unfair.

2. Second, based on Table 5, the paper argues that the ATSL loss proposed by the authors is superior to other loss functions. Nevertheless, the gain brought by ATSL loss is minimal—this gain may likely come from the added cumbersome computational process, and the effect is not significant.

3. Finally, the experiments presented in Tables 7 and 8 use non-state-of-the-art baselines, which is unreasonable.

**Questions:**

See Weaknesses.

---

> ### Author Response · Authors · 2025-11-25
> **Author Response to Reviewer 9XJh (Weakness 1)**
>
> We sincerely thank the Reviewer for the thoughtful and constructive feedback. We deeply appreciate the positive evaluation of our writing, logic, motivation and methodology. Below, we address your concerns in detail.
>
> Moreover, we have incorporated relevant analyses and experiments into the main body and appendix **in the revised manuscript** (where **modifications** to the original content are **marked in light purple**, and **new additions are highlighted in green**).
>
> ---
>
> > W1: First, the paper points out that MD-RE will improve the model's performance on the DocRE task. However, MD-RE adopts multiple discriminators, and the additional computational and time costs incurred have not been evaluated. This makes comparisons with previous methods unfair.
>
> Thank you for your insightful comment. **We would like to clarify that the computational and time costs of MD-RE have already been evaluated in original Appendix C.1**. *These details have now been moved to Section 6.3 (Resource Efficiency, Q7) in the main body of revised manuscript.*
>
> As shown in **Table 1 below** (i.e., Table 9 in Section 6.3 of the revised manuscript), MD-RE achieves a balance between performance and efficiency. While achieving a state-of-the-art F1 score of **77.80**, MD-RE maintains a modest computational cost. Specifically:
>
> 1. **Memory:** It requires only **17.63 GiB** of memory, which is significantly lower than TTM-RE (20.30 GiB) and less than half the memory consumption of evidence-based methods like Eider and SAIS (>43 GiB).
> 2. **Training Time:** Its training time (**81.80 min**) is substantially faster than KD-DocRE and TTM-RE, reducing the time cost by over 50%.
>
> Although MD-RE incurs a marginal increase in resource usage compared to lightweight models like ATLOP or DREEAM, this trade-off yields a substantial performance gain (+3.57 F1 over DREEAM), demonstrating its effectiveness without incurring excessive computational costs. **The overall results demonstrate that MD-RE improves the model's performance on the DocRE task, while also being efficient in terms of computational resources.**
>
> ***Table 1: Resource usage under BERT$_{base}$ on Re-DocRED (batch size 4).***
>
> |Method|Memory (GiB)|Training Time (Min)|F1|
> |-|-|-|-|
> |**Without evidence**||||
> |ATLOP|10.37|55.80|74.02|
> |KD-DocRE|14.71|169.59|74.55|
> |TTM-RE|20.30|208.37|-|
> |**With evidence**||||
> |Eider|43.10|-|-|
> |SAIS|46.20|-|-|
> |DREEAM|13.46|59.02|74.23|
> |**MD-RE (ours)**|17.63|81.80|**77.80**|
> |

---

> ### Author Response · Authors · 2025-11-25
> **Author Response to Reviewer 9XJh (continued, Weakness 2)**
>
> > W2: Second, based on Table 5, the paper argues that the ATSL loss proposed by the authors is superior to other loss functions. Nevertheless, the gain brought by ATSL loss is minimal—this gain may likely come from the added cumbersome computational process, and the effect is not significant.
>
> Thank you very much for your valuable comment. We would like to address your concern about our ATLS loss by providing further clarification in two aspects:
>
> 1. **The gain is not due to the added computational process**: We compared the training times of different losses, as shown in **new Table 2 below** (*which is included in Section 6.4 of the revised manuscript*). The training time for ATSL is 57.12 minutes, which is comparable to, and in many cases faster than, other baseline loss functions. For instance, ATSL is faster than standard ATL (59.71 min) and HingeABL (59.87 min), and it achieves state-of-the-art extraction performance. These results confirm that the performance gain is not due to an "added cumbersome computational process".
>
> ***Table 2: Training time comparison of ATSL and other losses on the ATLOP backbone with BERT$_{\text{base}}$, trained for 30 epochs on Re-DocRED (batch size 4). Training times may slightly vary due to the saving of model checkpoints.***
>
> |Loss Function|Approx. Training Time|
> |-|-|
> |ATL|59.71 minutes|
> |Balanced-Softmax|57.27 minutes|
> |AML|54.50 minutes|
> |AFL|60.45 minutes|
> |HingeABL$_{SAT}$|58.39 minutes|
> |HingeABL$_{MeanSAT}$|60.81 minutes|
> |HingeABL|59.87 minutes|
> |CMM|- (No available code)|
> |**ATSL (Ours)**|57.12 minutes|
> |
>
> 2. **The performance gains from ATSL are substantial and robust**:
>
> - First, comparing with other losses (Table 5 in the initial submission), ATSL improves performance over the main baseline ATLOP, boosting the F1 score from 73.29 to 76.48 (a **+3.19** F1 gain). It also outperforms HingeABL by **+1.33** F1 and slightly improves on the recent CMM loss (**+0.36** F1).
> - Second, when replacing the loss in different models with our ATSL loss (Table 4 in the initial submission), ATSL consistently increases F1 scores by an average of **2.52**, demonstrating its broad applicability.
> - Lastly, we **conduct additional experiments** by analyzing False Negative (FN) rates and applying ATSL to multiple baseline models and include AUC as an evaluation metric. The **Table 3 below** (*which is included in Table 8 of Section 6.2 in the revised manuscript*) shows that ATSL effectively reduces FN and increases AUC across various baseline models, with an average AUC improvement of **4.53**.
>
> These combined results highlight the effectiveness and robustness of the ATSL loss, with performance gains that do not come from added computational cost.
>
> ***Table 3: Experimental results analyzing False Negative (FN) rates and applying ATSL to multiple baseline models, with AUC as an evaluation metric on the Re-DocRED dev set. FN (False Negative): Predicts a positive example as negative. FP (False Positive): Predicts a negative example as positive. FN_NA: Predicts a positive example as negative, and the predicted label is NA.***
>
> |Model|FN↓|FP↓|FN_NA|FN/(FN+FP)|AUC↑|
> |-|-|-|-|-|-|
> |**Re-DocRED with BERT$_{base}$**||||||
> |ATLOP **(2021)**|5833|1942|5054|0.75|60.81|
> |ATLOP with ATSL (ours)|4181|4029|3467|0.51|**67.66(+6.85↑)**|
> |DocuNet **(2021)**|5777|1955|4971|0.75|63.90|
> |DocuNet with ATSL (ours)|4111|4154|3428|0.50|**68.04(+4.14↑)**|
> |TTM-RE **(2024)**|4413|3745|3724|0.54|68.05|
> |TTM-RE with ATSL (ours)|3710|3107|2898|0.54|**73.55(+5.50↑)**|
> |VaeDiff-DocRE **(2025)**|5180|2525|4358|0.67|66.28|
> |VaeDiff-DocRE with ATSL (ours)|4313|3354|3664|0.56|**69.18(+2.90↑)**|
> |
> |**Re-DocRED with RoBERTa$_{large}$**||||||
> |ATLOP **(2021)**|5205|1741|4405|0.75|65.61|
> |ATLOP with ATSL (ours)|3695|2951|3052|0.56|**72.50(+6.89↑)**|
> |DocuNet **(2021)**|5126|1833|4203|0.74|67.73|
> |DocuNet with ATSL (ours)|4069|2592|3362|0.61|**70.32(+2.59↑)**|
> |TTM-RE **(2024)**|4245|2526|3504|0.63|70.04|
> |TTM-RE with ATSL (ours)|3766|2156|3020|0.64|**74.09(+4.05↑)**|
> |VaeDiff-DocRE **(2025)**|4974|1874|4099|0.73|68.12|
> |VaeDiff-DocRE with ATSL (ours)|4146|2636|3467|0.61|**71.46(+3.34↑)**|
> |

---

> ### Author Response · Authors · 2025-11-25
> **Author Response to Reviewer 9XJh (continued, Weakness 3)**
>
> > W3: Finally, the experiments presented in Tables 7 and 8 use non-state-of-the-art baselines, which is unreasonable.
>
>
> Thank you for your valuable comment. In the original submission, the analysis experiments presented in Tables 7 and 8 primarily focused on comparing the performance with the main baseline model, ATLOP. **To address your concern, we further evaluate our method against the latest 2024 and 2025 baselines, and update Tables 7 and 8 in the revised manuscript**.
>
> First, as shown in **Table 4 below** (i.e., *the revised Table 7 in Section 6.1 of the updated manuscript*),  applying our MD-RE framework consistently improves F1 scores for these baselines, **with an average gain of 2.23 F1**. These consistent improvements across losses demonstrate MD-RE’s effectiveness and generalizability.
>
> Moreover, we extend our class imbalance experiments and unapte Table 8 in Section 6.2 (i.e., the Table 3 of our response to your previous comment **W2**). We include the 2024 and 2025 baselines and **add AUC as an evaluation metric**. Applying ATSL across these models reduces false negatives and **increases AUC, with an average improvement of 4.53**. These improvements suggest that ATSL not only reduces false negatives but also improves overall discriminative ability and robustness under class imbalance.
>
> ***Table 4: Ours framework MD-RE vs. baselines under same losses, all using BERT$_{\text{base}}$ on the Re-DocRED test set (i.e., the revised Table 7 in the updated manuscript).***
>
> |Loss + Model|F1|Ign-F1|
> |-|-|-|
> |**ATL loss (2021)**|||
> |+ ATLOP|74.02|73.22|
> |+ MD-RE (ours)|**77.06 (+3.04↑)**|**76.02 (+2.80↑)**|
> |**AFL loss (2022)**|||
> |+ KD-DocRE|74.55|73.67|
> |+ MD-RE (ours)|**77.41 (+2.86↑)**|**76.19 (+2.52↑)**|
> |**NCRL loss (2024)**|||
> |+ ABRE|76.30|75.70|
> |+ MD-RE (ours)|**76.99 (+0.69↑)**|**75.97 (+0.27↑)**|
> |**PEMSCL loss (2025)**|||
> |+ VaeDiff-DocRE|75.07|74.13|
> |+ MD-RE (ours)|**77.40 (+2.33↑)**|**76.09 (+1.96↑)**|
> |
>
> ---
>
> Once again, thank you very much for all of your insightful comments. We truly hope that our detailed responses and the revisions made to the manuscript have addressed all your concerns.

---

### Official Review · Reviewer_wM2Y · 2025-10-26

**Soundness:** 3
**Presentation:** 2
**Contribution:** 3
**Rating:** 6
**Confidence:** 4

**Summary:**

MD-RE effectively addresses the issue of evidence-sentence dependency and reduces document noise through multi-discriminator fusion, showing solid performance across several datasets. ATSL’s threshold bias design mitigates class imbalance, particularly reducing false negatives. However, the paper lacks a direct comparison with multi-stage baseline models, and there is no analysis of how to mitigate ATSL’s increase in false positives. Additionally, MD-RE’s performance in long-document or low-resource scenarios remains untested.

**Strengths:**

1.The introduction of MD-RE addresses the limitation of evidence-sentence dependency by using multiple discriminators, which enhances the model's ability to handle document noise and supports multi-perspective reasoning.

2.ATSL's threshold bias design allows for flexible decision boundary adjustments, improving performance by reducing false negatives.

3.The extensive experiments demonstrate that MD-RE outperforms state-of-the-art methods across multiple datasets and shows generalization across various baselines.

**Weaknesses:**

1.No direct comparison with multi-stage or multi-discriminator DocRE models (e.g., hierarchical filtering methods) to clarify MD-RE’s novelty in framework design.

2.MD-RE’s performance on long documents (e.g., >10 sentences) or low-resource settings (e.g., few-shot) is untested, leaving its generalizability to challenging scenarios unclear.

3.The fusion module’s weighted sum strategy lacks ablation against other fusion methods (e.g., attention-based fusion), and the weighting factor α’s tuning rationale across datasets is not explained.

4.The class imbalance mitigation comes at an unacceptable cost of false positives. While ATSL reduces false negatives (FN) from 5833 to 4181, it triples false positives (FP) from 1942 to 4029 (Table 8). The paper does not explain how this FP surge is justified for real-world applications (e.g., knowledge graph construction, where FPs corrupt downstream tasks). This critical trade-off is ignored, undermining ATSL’s practical value.

5.The ATSL loss’s theoretical analysis is incomplete and inconsistent. The paper claims ATSL is "convex" but does not address that the convexity holds only for fixed P_T and N_T—in practice, these sets change dynamically with logits, making the overall loss non-convex.

6.No analysis of MD-RE’s failure cases beyond the single case study; it is unknown which relation types (e.g., long-tail) or document structures lead to underperformance.

**Questions:**

See Weaknesses.

---

> ### Author Response · Authors · 2025-11-25
> **Author Response to Reviewer wM2Y (Weakness 1,2)**
>
> We sincerely thank the Reviewer for the thoughtful and constructive feedback. We deeply appreciate the positive evaluation of our methodology and experiments. Below, we address your concerns in detail.
>
> Moreover, we have incorporated relevant analyses and experiments into the main body and appendix **in the revised manuscript** (where **modifications** to the original content are **marked in light purple**, and **new additions are highlighted in green**).
>
> ---
>
> > W1: No direct comparison with multi-stage or multi-discriminator DocRE models (e.g., hierarchical filtering methods) to clarify MD-RE’s novelty in framework design.
>
> Thank you for your valuable comment. We would like to clarify a possible misunderstanding. To the best of our knowledge, **there are currently no existing “multi-stage or multi-discriminator” frameworks specifically designed for DocRE**. Our proposed MD-RE is, to the best of our knowledge, **the first approach that introduces a multi-discriminator design into DocRE**.
>
> > W2: MD-RE’s performance on long documents (e.g., >10 sentences) or low-resource settings (e.g., few-shot) is untested, leaving its generalizability to challenging scenarios unclear.
>
> Thank you for your insightful comment. Following your suggestion, **we conduct new experiments on MD-RE’s performance with long documents (e.g., >10 sentences).** In detail, we first count the number of long documents in the Re-DocRED and DWIE datasets (**see new Table 1 below**). To evaluate MD-RE on long documents, we test it on Re-DocRED samples containing more than 10 sentences (**see new Table 2 below**). The results show that MD-RE achieves an **average improvement of 2.8–4.4 F1 on long documents** over strong baselines across different model. Moreover, we observed that the results in Section 5.1 of the initial submission, on long documents, which account for approximately 90% of the DWIE dataset, also demonstrate MD-RE’s advantage on long documents.
>
> Moreover, in Section 5.1 of our initial submission version, we conducted the experiment on the **low-resource setting**, where we trained MD-RE on the partially annotated DocRED dataset and evaluated it on the fully annotated Re-DocRED dataset. The results show that MD-RE surpasses strong baselines such as P<sup>3</sup>M and SSR-PU by 1.59–6.43 F1, **confirming its robustness and generalization in the low-resource setting**.
>
> *These further experiments (i.e., the new Table 1 and 2 below) are included in the revised manuscript (i.e., the Appendix B.2 and the Appendix D.1).*
>
> ***Table 1: Statistics of document lengths in Re-DocRED and DWIE datasets. Long documents are defined as those with more than 10 sentences.***
>
> |Dataset / Split|Total Docs|Long Docs (>10 sents)|Long / Total|
> |-|-|-|-|
> |**Re-DocRED Dataset**|
> |Train|3,053|499|16.34%|
> |Dev|500|87|17.40%|
> |Test|500|78|15.60%|
> |**DWIE  Dataset**|
> |Train|602|553|91.86%|
> |Dev|98|87|88.78%|
> |Test|99|92|92.93%|
> |
>
> ***Table 2: Results of the MD-RE framework evaluated on documents longer than 10 sentences.***
>
> |Model|Dev F1|Test F1|
> |-|-|-|
> |**Re-DocRED with BERT$_{base}$**|||
> |ATLOP (2021)|70.44|73.67|
> |DocuNet (2021)|70.17|71.81|
> |TTM-RE (2024)|71.93|73.01|
> |VaeDiff-DocRE (2025)|70.27|73.38|
> |**MD-RE (ours)**|**73.54**|**77.32**|
> |**Re-DocRED with RoBERTa$_{large}$**|
> |ATLOP (2021)|73.92|78.36|
> |DocuNet (2021)|75.01|76.54|
> |TTM-RE (2024)|77.52|79.96|
> |VaeDiff-DocRE (2025)|74.47|77.09|
> |**MD-RE (ours)**|**78.07**|**80.76**|
> |

---

> ### Author Response · Authors · 2025-11-25
> **Author Response to Reviewer wM2Y (continued, Weakness 3,4)**
>
> > W3: The fusion module’s weighted sum strategy lacks ablation against other fusion methods (e.g., attention-based fusion), and the weighting factor α’s tuning rationale across datasets is not explained.
>
> Thank you for your insightful comment. We apologize for the ambiguity in the terminology used in the ablation studies of the original submission, which may have made the fusion strategy comparisons less clear. **The ablation studies of Table 6 in the original manuscript included comparisons between our weighted fusion and several alternative fusion strategies**, though the naming in Table 6 may have caused confusion. In the initial submission version, “fusion” referred to weighted fusion, while “Add” and “Pipeline” denoted alternative fusion strategies. To avoid ambiguity, the revised manuscript updates these names to “Weighted” fusion, “Add” fusion, and “Pipeline” fusion, respectively. To further strengthen the comparison, we also **add two additional variants**, “Mean" fusion and “Voting" fusion. **Note that we do not include attention-based fusion, as our fusion occurs only during reasoning and does not involve model training, making attention-based fusion unsuitable in this setting**.
>
> The results on the Re-DocRED dev set using BERT$_{base}$ are shown in **Table 3 below**. **Our weighted fusion consistently achieves the best performance**, demonstrating its advantage over alternative fusion strategies.
>
> *The original Table 6 is updated and these further experiments are included in the revised manuscript (i.e., Table 15 in Appendix C).*
>
> ***Table 3: Ablation study of fusion strategies on the Re-DocRED dev set using BERT$_{base}$ as the encoder.***
>
> |Model|F1|Ign-F1|
> |-|-|-|
> |**MD-RE (ours)**|||
> |`with` Weighted Fusion|**77.70**|**76.46**|
> |`with` Pipeline Fusion|72.83|72.27|
> |`with` Add Fusion|77.02|75.53|
> |`with` Mean Fusion (New)|76.05|75.21|
> |`with` Voting Fusion  (New)|72.63|72.08|
> |
>
> In addition, **we would like to clarify that we have already conducted and reported a detailed study on the α hyperparameter in Appendix D.4**, where we analyze its effect on Re-DocRED and DWIE. The results show a consistent trend: performance increases with α until a peak and then declines. The optimal value lies **within a stable range** (0.6–0.8 for Re-DocRED and 1.0–1.2 for DWIE), with performance changes within <1.5 F1, indicating that the method is not highly sensitive to α. Based on these findings, we recommend **α = 1.0 as a default**. We have made this guidance more explicit in the revised version.
>
> > W4: The class imbalance mitigation comes at an unacceptable cost ...... FP surge is justified for real-world applications (e.g., knowledge graph construction, where FPs corrupt downstream tasks). This critical trade-off is ignored, undermining ATSL’s practical value.
>
> We thank the reviewer for highlighting the trade-off between FN and FP in ATSL. To better illustrate this effect, **we conduct additional experiments applying ATSL to multiple baseline models and include AUC as an evaluation metric**. As shown in **Table 4 below**, ATSL consistently enhances AUC by 2.59–6.89 points across all models, indicating that it strengthens the model’s capacity to correctly identify positive instances even with an increase in false positives.  From the results, we can conclude that our ATLS overall improves document extraction performance, which is consistent with the results from our other detailed comparison experiments in Sections 5 and 6, and in turn, benefits downstream tasks.
>
> *These further experiments in Table 4 below are included in the revised manuscript (i.e., Table 8 in Section 6.2 of the main body).*
>
> ***Table 4: Experimental results on class imbalance on Re-DocRED dev set. FN (False Negative): Predicts a positive example as negative. FP (False Positive): Predicts a negative example as positive. FN_NA: Predicts a positive example as negative, and the predicted label is NA.***
>
> |Model|FN↓|FP↓|FN_NA|FN/(FN+FP)|AUC↑|
> |-|-|-|-|-|-|
> |**Re-DocRED with BERT$_{base}$**||||||
> |ATLOP **(2021)**|5833|1942|5054|0.75|60.81|
> |ATLOP with ATSL (ours)|4181|4029|3467|0.51|**67.66(+6.85↑)**|
> |DocuNet **(2021)**|5777|1955|4971|0.75|63.90|
> |DocuNet with ATSL (ours)|4111|4154|3428|0.50|**68.04(+4.14↑)**|
> |TTM-RE **(2024)**|4413|3745|3724|0.54|68.05|
> |TTM-RE with ATSL (ours)|3710|3107|2898|0.54|**73.55(+5.50↑)**|
> |VaeDiff-DocRE **(2025)**|5180|2525|4358|0.67|66.28|
> |VaeDiff-DocRE with ATSL (ours)|4313|3354|3664|0.56|**69.18(+2.90↑)**|
> |
> |**Re-DocRED with RoBERTa$_{large}$**||||||
> |ATLOP **(2021)**|5205|1741|4405|0.75|65.61|
> |ATLOP with ATSL (ours)|3695|2951|3052|0.56|**72.50(+6.89↑)**|
> |DocuNet **(2021)**|5126|1833|4203|0.74|67.73|
> |DocuNet with ATSL (ours)|4069|2592|3362|0.61|**70.32(+2.59↑)**|
> |TTM-RE **(2024)**|4245|2526|3504|0.63|70.04|
> |TTM-RE with ATSL (ours)|3766|2156|3020|0.64|**74.09(+4.05↑)**|
> |VaeDiff-DocRE **(2025)**|4974|1874|4099|0.73|68.12|
> |VaeDiff-DocRE with ATSL (ours)|4146|2636|3467|0.61|**71.46(+3.34↑)**|
> |

---

> ### Author Response · Authors · 2025-11-25
> **Author Response to Reviewer wM2Y (continued, Weakness 5)**
>
> > W5: The ATSL loss’s theoretical analysis is incomplete and inconsistent. The paper claims ATSL is "convex" but does not address that the convexity holds only for fixed P_T and N_T—in practice, these sets change dynamically with logits, making the overall loss non-convex.
>
> We sincerely thank the reviewer for insightful observation. We acknowledge that the notation used in the **original Appendix A.1 was ambiguous, which led to the misunderstanding that the sets $\mathcal{P}_T$ and $\mathcal{N}_T$ depend on the model's predictions (logits)**.
>
> 1. **Clarification on Definition (Ground-Truth vs. Prediction)**
>
> We would like to clarify that in the context of the ATSL loss calculation and its optimization (as intended in our method and code), $\mathcal{P}_T$ and $\mathcal{N}_T$ strictly denote the subsets of ground-truth relations for a given entity pair, independent of the current model predictions.
>
> Specifically, the definitions of $\mathcal{P}_T$ and $\mathcal{N}_T$ in the theoretical analysis of Appendix  A.1 are revised as follows:
>
> - $\mathcal{P}_T = \{r \in R \mid y_r = 1\}$ (Relations actually expressed by an entity pair)
> - $\mathcal{N}_T = \{r \in R \mid y_r = 0\}$ (Relations not expressed)
>
> Under this standard supervised learning formulation, for any given input sample $x$ and its labels $y$, the sets $\mathcal{P}_T$ and $\mathcal{N}_T$ are **constant** with respect to the logits.
>
> 2. **Re-validation of Convexity**
>
> With $\mathcal{P}_T$ and $\mathcal{N}_T$ being fixed sets determined by the ground truth:
>
> - The term $\mathcal{L}_1'$ involves the Log-Sum-Exp function over fixed indices in $\mathcal{P}_T$, minus a linear term.
>
> - The term $\mathcal{L}_2'$ involves the Log-Sum-Exp function over fixed indices in $\mathcal{N}_T$, minus a linear term.
>
>   Since the Log-Sum-Exp function is convex and convexity is preserved under affine transformations, the total loss $\mathcal{L}_{ATSL}$ is strictly convex with respect to the logits. This ensures that the optimization landscape is well-behaved.
>
> We have revised Theoretical Analysis of ATSL Loss (Appendix A in the revised manuscript) to align the notation strictly with the Problem Formulation (Section 3.1) in the main text.

---

> ### Author Response · Authors · 2025-11-25
> **Author Response to Reviewer wM2Y (continued, Weakness 6)**
>
> > W6: No analysis of MD-RE’s failure cases beyond the single case study; it is unknown which relation types (e.g., long-tail) or document structures lead to underperformance.
>
> Thank you for your insightful comment. Following your suggestions, we **first conduct additional experiments to analyze relation-level performance on Re-DocRED (see new Table 5 below)** and observe that MD-RE performs strongly on frequent relations (F1 > 80) but drops on long-tail relations (e.g., *replaced by*: 22.22; *replaces*: 12.50), indicating that most errors arise from sparsely represented relations. False positives occur more often in high-frequency relations, suggesting occasional over-prediction.
>
> To further evaluate **long-tail** robustness, we **additionally report** performance separately on frequent (Freq) and long-tail (LTail) subsets (**see new Table 6 below**). MD-RE consistently improves long-tail F1 ($\text{BERT}{\text{base}}$: +2.50 dev / +3.02 test; $\text{RoBERTa}{\text{large}}$: +2.67 dev / +2.60 test) while maintaining frequent-relation performance, demonstrating robust modeling of sparse relations and overall generalization.
>
> *These further experiments (new Table 5 and 6 below) are included in Appendix D.5.2 of the revised manuscript.*
>
> ***Table 5: Head vs. Tail relation performance of our MD-RE framework on the Re-DocRED dev set with BERT$_{base}$.***
>
> |Relation name|FP|FN|Ture samples|F1|Rank (by train set, descending)|
> |-|-|-|-|-|-|
> |located in the administrative territorial entity|635|696|3846|82.56|1|
> |country|414|449|2739|84.14|2|
> |country of citizenship|194|147|823|79.86|3|
> |contains administrative territorial entity|114|155|662|79.03|4|
> |notable work|82|127|604|82.03|5|
> |has part|75|196|469|66.83|6|
> |part of|77|192|444|65.20|7|
> |||||||
> |manufacturer|15|20|76|76.19|80|
> |parent taxon|3|5|28|85.19|81|
> |original language of work|3|9|19|62.50|82|
> |end time|10|16|26|43.48|83|
> |location of formation|1|9|24|75.00|84|
> |replaced by|3|11|13|22.22|85|
> |replaces|2|12|13|12.50|86|
> |narrative location|2|1|7|80.00|87|
> |
>
> ***Table 6: Long-Tail relation performance with different models. Experimental result on Re-DocRED dataset. Freq F1 evaluates the 10 most common relations, while LTail F1 covers the remaining 86 uncommon relations in Re-DocRED.***
>
> ||**Dev Set**||||**Test Set**||||
> |:-|:-:|:-:|:-:|:-:|:-:|:-:|:-:|:-:|
> |**Model**|**F1**|**Ign F1**|**Freq F1**|**LTail F1**|**F1**|**Ign F1**|**Freq F1**|**LTail F1**|
> |**BERT$_{\text{base}}$**|||||||||
> |ATLOP|74.22|73.35|77.77|68.83|74.02|73.22|75.92|67.46|
> |DocuNET|74.65|73.68|78.55|69.41|74.49|73.60|77.54|68.33|
> |KD-DocRE|74.69|73.76|78.71|70.26|74.55|73.67|78.28|69.52|
> |TTM-RE|76.21|74.74|80.45|69.42|76.33|74.89|79.83|68.99|
> |VaeDiff-DocRE|75.89|74.96|79.18|71.12|75.07|74.13|78.32|70.06|
> |**MD-RE (ours)**|**77.70 (1.49↑)**|**76.46 (1.50↑)**|**80.51 (0.06↑)**|**73.62 (2.50↑)**|**77.80 (1.47↑)**|**76.63 (1.74↑)**|**80.81 (0.98↑)**|**73.18 (3.02↑)**|
> |**RoBERTa$_{\text{large}}$**|||||||||
> |ATLOP|77.63|76.88|80.92|72.83|77.73|76.94|80.78|72.29|
> |DocuNET|78.16|77.53|80.73|73.34|77.92|77.27|81.25|73.32|
> |KD-DocRE|78.65|77.92|80.52|72.54|78.35|77.63|80.85|74.31|
> |TTM-RE|78.13|78.05|83.05|73.90|79.95|78.20|83.43|73.23|
> |VaeDiff-DocRE|79.19|78.35|82.00|75.13|79.03|78.22|81.84|74.73|
> |**MD-RE (ours)**|**81.44 (2.25↑)**|**80.38 (2.03↑)**|**84.10 (1.05↑)**|**77.80 (2.67↑)**|**81.49 (1.54↑)**|**80.45 (2.23↑)**|**84.10 (0.67↑)**|**77.33 (2.60↑)**|
> |
>
> ---
> Once again, thank you very much for all of your insightful comments. We truly hope that our detailed responses and the revisions made to the manuscript have addressed all your concerns.

---

### Official Review · Reviewer_fT89 · 2025-11-01

**Soundness:** 3
**Presentation:** 4
**Contribution:** 3
**Rating:** 6
**Confidence:** 3

**Summary:**

This paper presents a document-level relation extraction method that integrates a multi-discriminator framework (MD-RE) with an Adaptive Threshold Shifted Loss (ATSL). The main innovation lies in its three-stage discriminator design (recall, coarse, and fine), which jointly addresses noise reduction and class imbalance without relying on external evidence sentences. The authors conducted comprehensive experiments on three benchmark datasets, demonstrating that the proposed approach achieves state-of-the-art performance on multiple metrics, while the ATSL loss exhibits strong generalization ability across different models.

**Strengths:**

1. The authors accurately identify a key challenge in DocRE — severe class imbalance that causes overly high adaptive thresholds and excessive false negatives — and propose ATSL as a simple yet generalizable solution.
2. By introducing bias terms (λ, β), ATSL mathematically redefines the margin constraints between positive/negative samples and decision thresholds. The convexity proof and Bayes-consistency analysis in the appendix provide a solid theoretical foundation for the method.
3. Extensive experiments across three mainstream datasets (including loss replacement, disabling LNS, and removing discriminators) consistently verify the effectiveness of both MD-RE and ATSL, and confirm the adaptability of ATSL to various baseline models.

**Weaknesses:**

1. Although Figure 2 outlines the overall framework, the differences among the three discriminators’ decision behaviors are not clearly visualized. For example, what specific error types are filtered by the coarse versus the fine discriminator? The lack of such case studies makes the internal working mechanism less intuitive.
2. The paper notes that the optimal α parameter varies across datasets but does not provide systematic tuning guidelines. This sensitivity could hinder real-world deployment, particularly in low-resource scenarios without a dedicated validation set.
3. LNS selects top-k negative samples based on loss values, yet no comparison with alternative strategies (e.g., random sampling, margin-based mining) is provided, and ablation studies suggest its contribution is marginal.
4. The final fusion module uses fixed weighted averaging (Eq. 7–8). Although ablation studies show its effectiveness, the absence of comparisons with other fusion strategies makes it difficult to judge the optimality of this design.

**Questions:**

1. What specific types of false positives increase after applying ATSL? Could rule-based post-processing help reduce them?
2. For documents with sparse relational structures, is the full three-stage filtering necessary? It would be valuable to evaluate a subset of DWIE with low relation density to test the method’s generalizability.
3. The rule “directly reject if the recall discriminator predicts NA” — does it occasionally remove true positives? If so, what is the approximate proportion?

---

> ### Author Response · Authors · 2025-11-25
> **Author Response to Reviewer fT89 (Weakness 1,2,3)**
>
> We sincerely thank the Reviewer for the thoughtful and constructive feedback. We deeply appreciate the positive evaluation of our writing, methodology, theory, and experiments. Below, we address your concerns in detail.
>
> Moreover, we have incorporated relevant analyses and experiments into the main body and appendix **in the revised manuscript** (where **modifications** to the original content are **marked in light purple**, and **new additions are highlighted in green**).
>
> ---
>
> > W1: Although Figure 2 outlines the overall framework, the differences among the three discriminators’ decision behaviors are not clearly visualized. For example, what specific error types are filtered by the coarse versus the fine discriminator? The lack of such case studies makes the internal working mechanism less intuitive.
>
> We sincerely thank the reviewer for the valuable comments. **We elaborate on this point from three perspectives**: (1) We fully agree that visualizing the differences in decision behaviors among the three discriminators makes their internal mechanism more intuitive. Following your suggestion, **Figure 2 is revised** to show how each discriminator handles different types of relations. (2) We further **conduct new analyses and provide concrete error type examples** in **new Table 1 below**. (3) The **Appendix D.5.1 of the paper also provided case studies and detailed analyses** illustrating the discriminators’ decision patterns.
>
> ***Table 1:  Decision outcomes of Recall, Coarse, and Fine discriminators. Results on the Re-DocRED dev set using BERT$_{base}$ as the encoder. TP: True Positive. FN (False Negative): Predicts a positive example as negative. FP (False Positive): Predicts a negative example as positive.***
>
> |Relation Name|TP|FP|FN|Precision|Recall|F1|
> |-|-|-|-|-|-|-|
> |**Recall Discriminator**|
> |date of death (P570)|144|24|14|85.71|91.14|88.34|
> |cast member (P161)|167|30|23|84.77|87.89|86.30|
> |screenwriter (P58)|32|8|5|80.00|86.49|83.12|
> |**Coarse Discriminator**|
> |educated at (P69)|76|7|9|91.57|89.41|90.48|
> |publication date (P577)|239|37|56|86.59|81.02|83.71|
> |country (P17)|2172|308|567|87.58|79.30|83.23|
> |**Fine Discriminator**|
> |place of birth (P19)|93|4|33|95.88|73.81|83.41|
> |performer (P175)|235|15|95|94.00|71.21|81.03|
> |director (P57)|48|4|16|92.31|75.00|82.76|
> |
>
> > W2: The paper notes that the optimal α parameter varies across datasets but does not provide systematic tuning guidelines. This sensitivity could hinder real-world deployment, particularly in low-resource scenarios without a dedicated validation set.
>
> Thank you for your valuable comment. **We would like to clarify that we have already conducted and reported a detailed study on the α hyperparameter in Appendix D.4**, where we analyze its effect on Re-DocRED and DWIE. The results show a consistent trend: performance increases with α until a peak and then declines. The optimal value lies **within a stable range** (0.6–0.8 for Re-DocRED and 1.0–1.2 for DWIE), with performance changes within <1.5 F1, indicating that the method is not highly sensitive to α. Based on these findings, we recommend **α = 1.0 as a default**. We have made this guidance more explicit in the revised version.
>
> > W3: LNS selects top-k negative samples based on loss values, yet no comparison with alternative strategies (e.g., random sampling, margin-based mining) is provided, and ablation studies suggest its contribution is marginal.
>
> Thank you for your insightful suggestion. To evaluate the effect of LNS, we **conduct additional ablation experiments (new Table 2 below)**. **Our LNS consistently achieves the best performance**, improving both F1 and Ign-F1 compared to random and margin-based sampling. These results indicate that, although the gains are modest, LNS can also work in conjunction with the ATSL loss to dynamically adjust the threshold and recall of each discriminator.  *These further experiments (Table 2 below) are included in the revised manuscript (i.e., Table 14 in Appendix C).*
>
> ***Table 2 : Ablation study of sample selection strategies on the Re-DocRED dev set using BERT$_{base}$ as the encoder.***
>
> |Model|F1|Ign-F1|
> |-|-|-|
> |**MD-RE (ours)**|||
> |`with` LNS|**77.70**|**76.46**|
> |`with` Random sampling|77.57|76.39|
> |`with` Margin-based mining|77.44|75.87|
> |

---

> ### Author Response · Authors · 2025-11-25
> **Author Response to Reviewer fT89 (continued, Weakness 4 and Q1)**
>
> > W4: The final fusion module uses fixed weighted averaging (Eq. 7–8). Although ablation studies show its effectiveness, the absence of comparisons with other fusion strategies makes it difficult to judge the optimality of this design.
>
> Thank you for your insightful comment. We apologize for the ambiguity in the terminology used in the ablation studies of the original submission, which may have made the fusion strategy comparisons less clear. **The ablation studies of Table 6 in the original manuscript included comparisons between our weighted fusion and several alternative fusion strategies**, though the naming in Table 6 may have caused confusion. In the initial submission version, “fusion” referred to weighted fusion, while “Add” and “Pipeline” denoted alternative fusion strategies. To avoid ambiguity, the revised manuscript updates these names to “Weighted” fusion, “Add” fusion, and “Pipeline” fusion, respectively.
>
> To further strengthen the comparison, we also **add two additional variants**, “Mean" fusion and “Voting" fusion. Their results on the Re-DocRED dev set using BERT$_{base}$ are shown in **Table 3 below**, where **our weighted fusion consistently achieves the best performance**, demonstrating its advantage over alternative fusion strategies.
>
> *The original Table 6 is updated and these further experiments are included in the revised manuscript (i.e., Table 15 in Appendix C).*
>
> ***Table 3 : Ablation study of fusion strategies on the Re-DocRED dev set using BERT$_{base}$ as the encoder.***
>
> |Model|F1|Ign-F1|
> |-|-|-|
> |**MD-RE (ours)**|||
> |`with` Weighted Fusion|**77.70**|**76.46**|
> |`with` Pipeline Fusion|72.83|72.27|
> |`with` Add Fusion|77.02|75.53|
> |`with` Mean Fusion (New)|76.05|75.21|
> |`with` Voting Fusion  (New)|72.63|72.08|
> |
>
> > Q1: What specific types of false positives increase after applying ATSL? Could rule-based post-processing help reduce them?
>
> Thank you for your insightful comment. To investigate which types of false positives (FP) increase after applying ATSL, we conduct **additional experiments** and analyze FP counts by relation type. The results in **new Table 4 below** show that the FP increase mainly occurs in high-frequency relations, such as `located in the administrative territorial entity` and `country`, whereas long-tail relations experience smaller FP growth. This suggests that ATSL tends to over-predict on frequency relations. While the simple rule-based post-processing (e.g., confidence thresholding or relation-specific constraints) could potentially mitigate these additional false positives, we leave this exploration to future work.  *These further experiments (Table 4 below) are included in the revised manuscript (i.e., Table 19 in the Appendix D.5.3).*
>
> ***Table 4: Analysis of increased false positives after applying ATSL on Re-DocRED dev set.***
>
> |Relation name|ATLOP (FP)|ATLOP with ATSL (FP)|ΔFP|Rank (by train set, descending)|
> |-|-|-|-|-|
> |**Re-DocRED with BERT$_{base}$**|||||
> |located in the administrative territorial entity|411|716|305|1|
> |country|283|449|166|2|
> |country of citizenship|135|222|87|3|
> |contains administrative territorial entity|57|143|86|4||part of|40|113|73|7|
> |applies to jurisdiction|28|79|51|11|
> |has part|53|104|51|6|
> |publication date|38|85|47|9|
> |member of|32|78|46|10|
> |owned by|11|53|42|38|
> |**Re-DocRED with RoBERTa$_{large}$**|
> |located in the administrative territorial entity|355|490|135|1|
> |country|255|346|91|2||part of|49|106|57|7|
> |continent|23|63|40|20|
> |has part|54|93|39|6|
> |contains administrative territorial entity|53|91|38|4|]
> |member of|23|60|37|10|
> |country of citizenship|124|158|34|3|
> |headquarters location|6|37|31|55|
> |country of origin|33|63|30|16|
> |

---

> ### Author Response · Authors · 2025-11-25
> **Author Response to Reviewer fT89 (continued, Q2,3)**
>
> > Q2: For documents with sparse relational structures, is the full three-stage filtering necessary? It would be valuable to evaluate a subset of DWIE with low relation density to test the method’s generalizability.
>
> Thank you for your insightful comment. Following your suggestion, **we conduct additional experiments on a low-relation-density subset of DWIE** to evaluate the generalizability of MD-RE. Specifically, we select the bottom 35% of documents ranked by the number of relations (**see new Table 5**). We then evaluate MD-RE with different numbers of discriminators on this subset (**see new Table 6**). The results show that using three discriminators achieves the best performance (F1 = 78.52), indicating that **the full three-stage filtering remains effective and stable even for documents with sparse relational structures**.
>
> ***Table 5: Statistics of DWIE and its low-relation-density subset. \* Bottom 35% of documents after sorting by the number of relations.***
>
> |Dataset / Split|\#Docs (Avg. relations per doc)|Low-relation-density subset*|
> |-|-|-|
> |**DWIE Dataset**|||
> |Train|602 (23.94)|210 (8.48)|
> |Dev|98 (26.78)|34 (8.12)|
> |Test|99 (24.84)|34 (7.94)|
> |
>
> ***Table 6: Performance on the Low-relation-density subset of the DWIE test set with varying numbers of discriminators using BERT$_{base}$.***
>
> |Discriminator number|2|3|4|
> |-|-|-|-|
> |F1|77.31|78.52|77.74|
>
> > Q3: The rule “directly reject if the recall discriminator predicts NA” — does it occasionally remove true positives? If so, what is the approximate proportion?
>
> Thank you for your valuable comment. **We conduct additional experiments on Re-DocRED** and find that while the rejection rule of NA occasionally removes true positives, the proportion is relatively low. As shown in **Table 7 below**, compared with baseline models, our Recall Discriminator achieves the lowest FN_NA rates, filtering uncertain predictions more effectively while minimizing true positive loss. Only **20.06%** ($\text{BERT}{\text{base}}$) and **17.66%** ($\text{RoBERTa}{\text{large}}$) of positives are incorrectly rejected.
>
> ***Table 7: Proportion of true positives (TP) incorrectly removed by the NA rejection rule on the Re-DocRED Dev Set. FN (False Negative): Predicts a positive example as negative. FN_NA: Predicts a positive example as negative, and the predicted label is NA.***
>
> |Model|FN↓|FN_NA|FN_NA/All_Positives↓|
> |-|-|-|-|
> |**Re-DocRED with BERT$_{base}$**||||
> |ATLOP (2021)|5833|5054|29.24|
> |DocuNet (2022)|5777|4971|28.76|
> |TTM-RE (2024)|4413|3724|21.55|
> |VaeDiff-DocRE (2025)|5180|4358|25.21|
> |Recall Discriminator **(ours)**|4181|3467|**20.06**|
> |**Re-DocRED with RoBERTa$_{large}$**|
> |ATLOP (2021)|5205|4405|25.49|
> |DocuNet (2022)|5126|4203|24.32|
> |TTM-RE (2024)|4245|3504|20.27|
> |VaeDiff-DocRE (2025)|4974|4099|23.72|
> |Recall Discriminator **(ours)**|3695|3052|**17.66**|
> |
>
> ---
> Once again, thank you very much for all of your insightful comments. We truly hope that our detailed responses and the revisions made to the manuscript have addressed all your concerns.

---

### Official Review · Reviewer_SXpe · 2025-11-03

**Soundness:** 3
**Presentation:** 4
**Contribution:** 2
**Rating:** 4
**Confidence:** 3

**Summary:**

This paper tackles the problem of document-level relation extraction (DocRE), where models must determine relations between entity pairs across long texts. Existing DocRE systems often suffer from two issues: (1) High false-negative rates due to extreme class imbalance, causing the model to learn an overly strict relation-prediction threshold. (2) Noise from full-document encoding, which makes it difficult to distinguish subtle positive relations from dominant negative patterns.

The authors propose MD-RE, a Multi-Discrimination framework consisting of three discriminators with different decision thresholds: a recall-oriented discriminator, a coarse one, and a fine one. Instead of relying on sentence-extraction or evidence mining, these discriminators independently evaluate candidate relations from different perspectives and are combined via a weighted fusion mechanism. They introduce Adaptive Threshold Shifted Loss (ATSL), which shifts the learned threshold logits to counteract the high FN rates, and significantly include some good true positive candidates inside the decision boundary.

**Strengths:**

The multi-discriminatory design introduced by the author(s) is an interesting one. It targets a specific limitation of other methods involving high false negatives. Their recall-based discriminator helps in correcting a significant number of false negatives.

In particular, the redesigning of Adaptive-Threshold Loss (ATL) seems to be a stronger contribution of the author(s) to me. Because of the class imbalance with too many false negatives in the predictions, they introduce a shifting parameter in the loss calculation which significantly improves the decision mechanism of the discriminators. They call it the ATSL loss. They also provide satisfactory result analysis to represent the strength of this loss in their paper. The improved performance proves that the logits produced from the training follow the precision-recall trend of the individual discriminators.

**Weaknesses:**

* Though unlike traditional ensemble methods, the discriminators share the encoder, the fusion of the discriminators seems to be a specialized utilization of an ensemble method to me. I expect the author(s) to emphasize on this during the rebuttal.

* The removal of coarse, and fine discriminators show a slight decrease in F1 (still higher than other SOTAs) (Table 6). I would like to see how the method performs without both of them, only with the recall Discriminator.

* The method naturally includes a lot of False Positives in the game (Table 8). The authors did not provide any result analysis using the AUC scores. As a result, I cannot assess how strong the model is to discriminate between a positive and negative relation. I would like to see their result analysis on AUC metrics too.

**Questions:**

To assess the performance gain of the coarse and fine discriminator more deeply, I would like to see how the recall-only discriminator performs.

Please refer to the suggestions in the ‘Weaknesses’ section.

---

> ### Author Response · Authors · 2025-11-25
> **Author Response to Reviewer SXpe (Weakness 1,2)**
>
> We sincerely thank the Reviewer for the thoughtful and constructive feedback. We deeply appreciate the positive evaluation of our methodology and experiments. Below, we address your concerns in detail.
>
> Moreover, we have incorporated relevant analyses and experiments into the main body and appendix **in the revised manuscript** (where **modifications** to the original content are **marked in light purple**, and **new additions are highlighted in green**).
>
> ---
>
> > W1: Though unlike traditional ensemble methods, the discriminators share the encoder, the fusion of the discriminators seems to be a specialized utilization of an ensemble method to me. I expect the author(s) to emphasize on this during the rebuttal.
>
> We thank the reviewer for the valuable comments. We provide a detailed explanation from three perspectives. **First**, MD-RE introduces three discriminators with **distinct decision criteria** for DocRE, **which has not been explored in prior work**. These discriminators have different objectives and thresholds, focusing on high recall, coarse filtering, and fine-grained discrimination. This allows the model to evaluate candidate relations from predefined heterogeneous views, rather than averaging or voting over homogeneous predictions, highlighting a fundamental conceptual difference in purpose and decision-making. **Second**, we use the recall discriminator for Loss-aware Negative Selection (LNS), and the selected samples are then passed to the coarse and fine discriminators. **Finally**, the proposed **Adaptive Threshold Shifted Loss (ATSL)** introduces bias parameters to dynamically adjust each discriminator’s thresholds and recall–precision balance, enabling flexible optimization of decision boundaries during training and inference, rather than relying on static post-hoc ensemble strategies.
>
> > W2: The removal of coarse, and fine discriminators show a slight decrease in F1 (still higher than other SOTAs) (Table 6). I would like to see how the method performs without both of them, only with the recall Discriminator.
>
> Thank you for your insightful comment. We conduct **additional comprehensive experiments** using **only the recall discriminator**, and the results are shown in **new Table 1 below**.
>
> With only the recall discriminator, most results are inferior to those obtained using two or three discriminators. **These results further demonstrate the advantages of our discriminator design.** The recall discriminator alone is not sufficient to effectively handle more difficult negative or ambiguous samples. In contrast, the coarse and fine discriminators apply progressively stricter, more refined criteria, enabling them to better handle these challenging samples. Notably, the complementary filtering effect of the three discriminators leads to the best performance.
>
> *These further experiments (Table 1 below) are included in the revised manuscript (i.e., the updated Table 6 in the main body and the Table 13 in Appendix C).*
>
> ***Table 1: Ablation study of discriminators on the Re-DocRED and DWIE dev sets.***
>
> |Model|F1|Ign-F1|
> |-|-|-|
> |MD-RE on Re-DocRED using BERT$_{base}$ (ours)|**77.70**|**76.46**|
> |&nbsp;`w/o` Coarse Discriminator|77.18|75.67|
> |&nbsp;`w/o` Fine Discriminator|76.38|74.58|
> |&nbsp;`w/o` Coarse & Fine Discriminators (`only` Recall Discriminator)|76.23|74.83|
> |
> |MD-RE on Re-DocRED using RoBERTa$_{large}$ (ours)|**81.44**|**80.38**|
> |&nbsp;`w/o` Coarse Discriminator|80.70|79.33|
> |&nbsp;`w/o` Fine Discriminator|80.79|79.39|
> |&nbsp;`w/o` Coarse & Fine Discriminators (`only` Recall Discriminator)|80.35|79.22|
> |
> |MD-RE on DWIE using BERT$_{base}$ (ours)|**73.81**|**68.37**|
> |`w/o` Coarse Discriminator|72.67|68.24|
> |`w/o` Fine Discriminator|72.39|67.70|
> |`w/o` Coarse & Fine Discriminators (`only` Recall Discriminator)|69.14|61.11|
> |
> |MD-RE on DWIE using RoBERTa$_{large}$ (ours)|**77.11**|**73.13**|
> |`w/o` Coarse Discriminator|76.37|72.35|
> |`w/o` Fine Discriminator|76.67|72.47|
> |`w/o` Coarse & Fine Discriminators (`only` Recall Discriminator)|76.18|70.64|
> |

---

> ### Author Response · Authors · 2025-11-25
> **Author Response to Reviewer SXpe (continued, Weakness 3)**
>
> > W3: The method naturally includes a lot of False Positives in the game (Table 8). The authors did not provide any result analysis using the AUC scores. As a result, I cannot assess how strong the model is to discriminate between a positive and negative relation. I would like to see their result analysis on AUC metrics too.
>
> Thank you for your insightful comment. We **add AUC as an evaluation metric and conduct a more comprehensive analysis across multiple models (Table 2 below)**. The results show that **ATSL reduces false negatives and increases AUC across models, with an average AUC improvement of 4.53**.  These results indicate that ATSL effectively enhances the model’s ability to distinguish positive and negative relations.  *These further experiments (Table 2 below) are included in the revised manuscript (i.e., the updated Table 8 in Section 6.2 of the main body).*
>
> ***Table 2: Experimental results on class imbalance on Re-DocRED dev set. FN (False Negative): Predicts a positive example as negative. FP (False Positive): Predicts a negative example as positive. FN_NA: Predicts a positive example as negative, and the predicted label is NA.***
>
> |Model|FN↓|FP↓|FN_NA|FN/(FN+FP)|AUC↑|
> |-|-|-|-|-|-|
> |**Re-DocRED with BERT$_{base}$**||||||
> |ATLOP **(2021)**|5833|1942|5054|0.75|60.81|
> |ATLOP with ATSL (ours)|4181|4029|3467|0.51|**67.66(+6.85↑)**|
> |DocuNet **(2021)**|5777|1955|4971|0.75|63.90|
> |DocuNet with ATSL (ours)|4111|4154|3428|0.50|**68.04(+4.14↑)**|
> |TTM-RE **(2024)**|4413|3745|3724|0.54|68.05|
> |TTM-RE with ATSL (ours)|3710|3107|2898|0.54|**73.55(+5.50↑)**|
> |VaeDiff-DocRE **(2025)**|5180|2525|4358|0.67|66.28|
> |VaeDiff-DocRE with ATSL (ours)|4313|3354|3664|0.56|**69.18(+2.90↑)**|
> |
> |**Re-DocRED with RoBERTa$_{large}$**||||||
> |ATLOP **(2021)**|5205|1741|4405|0.75|65.61|
> |ATLOP with ATSL (ours)|3695|2951|3052|0.56|**72.50(+6.89↑)**|
> |DocuNet **(2021)**|5126|1833|4203|0.74|67.73|
> |DocuNet with ATSL (ours)|4069|2592|3362|0.61|**70.32(+2.59↑)**|
> |TTM-RE **(2024)**|4245|2526|3504|0.63|70.04|
> |TTM-RE with ATSL (ours)|3766|2156|3020|0.64|**74.09(+4.05↑)**|
> |VaeDiff-DocRE **(2025)**|4974|1874|4099|0.73|68.12|
> |VaeDiff-DocRE with ATSL (ours)|4146|2636|3467|0.61|**71.46(+3.34↑)**|
>
> ---
> Once again, thank you very much for all of your insightful comments. We truly hope that our detailed responses and the revisions made to the manuscript have addressed all your concerns.

---

### Author Response · Authors · 2025-11-30
**Summary of Reviewers‘ Comments and Our Responses**

We sincerely thank the AC for handling and reviewing our paper and the reviewers for their time, effort, and recognition of our work’s innovation, contributions, performance, generalizability, and clarity.

---

1. **Reviewers consistently affirm the `innovation and contributions` of our work**, particularly the introduction of *MD-RE* and *ATSL* for addressing document noise and extreme class imbalance.
   * Reviewer **SXpe** highlights *“interesting multi-discriminatory design…”* and that the redesign of ATL is *“…a stronger contribution.”*
   * Reviewer **fT89** emphasizes *“…the main innovation lies in its three-stage discriminator design,”* and recognized ATSL as *“a simple yet generalizable solution… provide a solid theoretical foundation.”*
   * Reviewer **wM2Y** notes that MD-RE *“…addresses the limitation of evidence-sentence dependency,”* and ATSL *“…alleviates the class imbalance.”*

2. **Reviewers consistently recognize the `strong performance and empirical validation`**, supported by extensive evaluation across multiple datasets and baselines.
   * Reviewer **SXpe** notes that *"...provide satisfactory result analysis to represent the strength..."*

   * Reviewer **fT89** comments that *“…extensive experiments across three mainstream datasets consistently verify the effectiveness.”*
   * Reviewer **wM2Y** states that *“…outperforms state-of-the-art methods across multiple datasets...and various baselines”*
   * Reviewer **9XJh** acknowledges *“The paper conducts extensive experiments...”*

3. **Reviewers  emphasize the `effectiveness and generalizability of ATSL`** in reducing false negatives under imbalance, showing solid theoretical and practical value.
   * Reviewer **SXpe** states *“the loss ... improves the decision mechanism.”*
   * Reviewer **wM2Y** states that *“The extensive experiments ... shows generalization across various baselines.”*
   * Reviewer **fT89** highlights *"ATSL as a simple yet generalizable solution."*

4. **Reviewers praise the `writing quality and clarity` of presentation.**
   * Reviewer **SXpe** and **fT89** both rate the presentation as *excellent (4)*.
   * Reviewer **9XJh** describes the paper as *“well-written, with clear logic.”*

&nbsp;

---

We provide a very careful and detailed rebuttal addressing the comments from all reviewers:

1. Reviewer **SXpe** raises questions regarding the performance of using *only Recall Discriminator* and *AUC results*.

   We **conduct detailed new experiments** on two aspects (Tables 1 and 2 in our response to W2 and W3). These additional experiments further show the effectiveness of our method (achieving an average AUC improvement of 4.53) .

   &nbsp;

2. Reviewer **fT89** raises questions regarding comparisons of *sampling strategies and fusion strategies*, how to adjust the α parameter, and additional *case studies*.

   **In the initial submission, we had already compared our method with several fusion strategies**. Furthermore, **we add** comparative experiments on **more** negative sampling and fusion strategies (Tables 2 and 3 in our response to W3 and W4). Results further show the advantages of our method over alternative strategies.

   Moreover, we have clearly indicated in Appendix D.4 of the initial submission that we already conducted a detailed analysis of the α parameter.

   Finally, in response to other questions, we also add new results, with most details included in Appendix D.5.

   &nbsp;

3. Reviewer **wM2Y** raises questions regarding the performance of our MD-RE on *long-document scenarios*, and analysis of failure *cases*.

   **We add long-document experiments** on both Re-DocRED and DWIE (Tables 1 and 2 in our response to W2). Results show that MD-RE achieves an **average improvement of 2.8–4.4 F1** over strong baselines across different models.

   We conduct a more systematic analysis of MD-RE failure cases, with detailed examples provided in our response to W6.

   &nbsp;

4. Reviewer **9XJh** raises questions regarding *resource usage analysis* for MD-RE and ATSL, and the addition of more baselines to supplement Tables 7 and 8.

   **In Appendix C.1 of our initial submission, we have reported the computational and time costs** of MD-RE, **now move to Section 6.3**.

   We also add ATSL resource usage experiments (in our response to W2), confirming that performance gains are achieved with reasonable resources.

   Finally, three additional baselines are included to supplement Tables 7 and 8 (in our response to W3).

---

We add the relevant analyses and experiments to the main body and appendix **in the revised manuscript (modifications in light purple, new additions in green)**, and `re-upload the revised manuscript`.

We hope our detailed responses and revisions address all reviewers’ suggestions and improve the overall quality of the paper. Once again, we sincerely thank the AC for your time and effort in handling and reviewing our work.

---

### Meta-Review · Area_Chair_eCtk · 2025-12-21

**Summary:**

The paper presents MD-RE, a multi-discrimination framework for document-level relation extraction with adaptive threshold shifted loss. The basic idea is to have three discriminators with different thresholds in order to make independent predictions. Then, the results are aggregated in a weighted fashion. In addition, the paper presents an adaptive threshold shifted loss to handle the class imbalance issue.

There are a number of major concerns regarding this paper:

1. The scope of interest is narrow, as the paper only focues on document-level relation extraction. It is unclear whether the framework can work well for other applications.

2. The paper has a couple of components, such as ensemble of discriminators and adaptive loss, but they're already present in previous work. Therefore, the novelty is limited.

3. Worst of all, the paper highly resembles a previously published paper: https://aclanthology.org/2025.findings-acl.1081.pdf
The resemblance includes (1) similar idea, (2) similar writing style, and (3) verbatim copy of certain text (at least 12 words in a row). Unfortunately, no citation is given.

Although the previous paper was published in August 2025, and this submission was in September 2025 and one may claim that there is not enough time for a thorough survey, the chance of the two papers being generated independently is extremely slim, especially considering verbatim copy of certain text.

If this paper has considered previous work (including ideas and writing), then the authors of this paper must cite previous work and properly credit the previous work, regardless of when the previous work was published. However, this is non-existing in the submission.

**Reviewer Concerns:**

The reviewers raised a number of concerns. Please see details of the reviews.

**Reviewer Scores:**

The authors provided certain responses, but they will not resurrect the paper.

---

### Decision · Program_Chairs · 2026-01-26

Reject